# S$^3$-NeRF: Neural Reflectance Field from Shading and Shadow under a Single Viewpoint

**Wenqi Yang**
The University of Hong Kong
`wqyang@cs.hku.hk`

**Guanying Chen**[*]
FNii and SSE, CUHK-Shenzhen
`chenguanying@cuhk.edu.cn`

**Chaofeng Chen**
Nanyang Technological University
`chaofenghust@gmail.com`

**Zhenfang Chen**
MIT-IBM Watson AI Lab
`chenzhenfang2013@gmail.com`

**Kwan-Yee K. Wong**
The University of Hong Kong
`kykwong@cs.hku.hk`

## Abstract

In this paper, we address the "dual problem" of multi-view scene reconstruction in which we utilize single-view images captured under different point lights to learn a neural scene representation. Different from existing single-view methods which can only recover a 2.5D scene representation (i.e., a normal / depth map for the visible surface), our method learns a neural reflectance field to represent the 3D geometry and BRDFs of a scene. Instead of relying on multi-view photo-consistency, our method exploits two information-rich monocular cues, namely shading and shadow, to infer scene geometry. Experiments on multiple challenging datasets show that our method is capable of recovering 3D geometry, including both visible and invisible parts, of a scene from single-view images. Thanks to the neural reflectance field representation, our method is robust to depth discontinuities. It supports applications like novel-view synthesis and relighting. Our code and model can be found at `https://ywq.github.io/s3nerf`.

## 1 Introduction

3D reconstruction from images is a central and important problem in computer vision. Multi-view stereo methods, which capture a target scene from multiple viewpoints under a fixed lighting condition [12, 24, 45, 46], are the most widely adopted approach for scene reconstruction. These methods, however, often assume surfaces with Lambertian reflectance and have difficulties in recovering high-frequency surface details.

An alternative approach to scene reconstruction is to utilize images captured from a fixed viewpoint but under different point light sources (see Fig. 1 (a)). This setup is adopted by photometric stereo (PS) methods [15, 47, 56] where shading information is utilized to reconstruct surface details of non-Lambertian objects. Shadow is another cue that has been exploited for shape recovery by shape-from-shadow methods [11, 61, 67]. However, existing single-view methods typically adopt a single normal or depth map to represent the visible surface, making them incapable of describing back-facing and occluded surfaces (see Fig. 1 (b)). Besides, methods relying on surface normal representation struggle

---

[*]Corresponding author

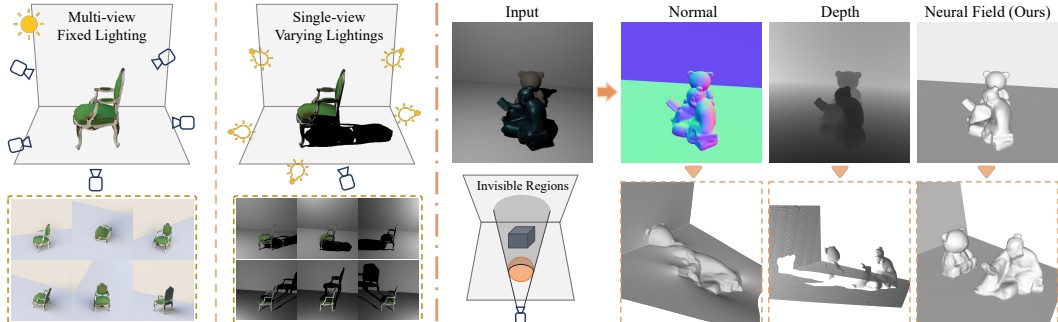

(a) Different capturing setups      (b) Comparison of different scene representations

Figure 1: (a) Difference between multi-view fixed lighting and single-view varying lighting setups. (b) Comparison of the normal map, depth map, and neural field in representing a 3D scene. Obtaining accurate depth from normal integration is non-trivial [5], and depth map cannot describe the invisible regions. The adopted neural field is capable of modeling the complete scene geometry.

to deal with depth discontinuities [25]. It is desirable to obtain a more complete scene reconstruction (including both visible and invisible parts) from single-view images. In this paper, we realize this by exploiting both shading and shadow cues to recover both visible and invisible parts of a scene.

Recently, neural scene representations have achieved significant progress in multi-view reconstruction and novel-view synthesis [35, 48, 63]. These methods model a continuous 3D space (i.e., the scene) with a multi-layer perception (MLP) which maps 3D points to scene properties (*e.g.*, density and color in NeRF [35]). Despite its great success in multi-view scene modeling, neural scene representation has been less explored in single-view scene modeling.

In this paper, building on top of the recent advances in neural scene representation, we propose to optimize a neural field using images captured from a single viewpoint under different point lights. Our method is fundamentally different from existing works [35, 48, 63] in that, instead of relying on multi-view photo-consistency, we exploit monocular shading and shadow cues to optimize our neural field for scene reconstruction (see Sec. A in supplementary for intuitive explanations on shadow cues).

A straightforward idea would be to condition the color MLP of NeRF [35] also on the point light directions. However, we find such a naïve solution fails to recover scene geometry and appearance. To make better use of the photometric stereo images, we explicitly model the surface geometry and BRDFs with a reflectance field and adopt a physics-based rendering to obtain the 3D point color [2, 3]. The 2D pixel color of a sampled ray can then be computed using volume rendering. Differentiable shadow computation is considered explicitly by tracing a ray from a 3D point to the point light position to check the light visibility [71]. As evaluating the light visibility of all points sampled along a ray is computationally expensive [49], we accelerate the computation by only evaluating the light visibility at the expected surface point, making online shadow computation possible during optimization.

To summarize, our contributions are:

- We address a novel problem of 3D neural reflectance field optimization from single-view images captured under different point lights. Different from existing neural scene representation methods that rely on multi-view photo-consistency, our method exploits monocular shading and shadow cues for neural field optimization.

- Our method jointly recovers the geometry and BRDFs of a scene, and adopts an efficient online shadow computation to fully exploit the information-rich shading and shadow cues.

- Experiments on multiple challenging datasets show that our method can faithfully reconstruct a complete scene geometry from single-view images. Our method is robust to depth discontinuities. It supports applications like novel-view synthesis and relighting.

## 2    Related Work

**Photometric stereo (PS)**   PS methods can recover pixel-wise surface normals from images captured under different light directions [15, 56]. Traditional PS methods treat specular observations as

outliers [36, 57, 58] or fit sophisticated reflectance models [10, 18, 54] to handle non-Lambertian surfaces. Recent methods resort to deep learning technique to solve this problem. Supervised learning methods learn a mapping from image observations to surface normals using synthetic dataset with ground-truth normals [7, 9, 17, 26, 32, 43, 73]. Self-supervised methods optimize the network parameters using an image reconstruction loss [22, 51]. The above methods assume directional lightings. For near-field PS problem, methods based on PDE [40] and deep learning [29, 31, 34, 44] have been proposed. More recently, Li *et al.* [25] proposes a coordinate-based MLP to represent the normal map of the visible surface assuming directional lights. In contrast, our method represents a scene with a continuous volume and recovers the full 3D scene geometry under a near-field setup.

**Shape from shadow**  Shadow has been exploited to estimate shape information [11]. Yu and Chang optimized a height map from shadow cues using a graph-based representation [67]. Shadowcuts [6] explicitly considers shadow in Lambertian photometric stereo. Yamashita *et al.* [61] introduced a 1D shadow graph to accelerate the shadow computation. Recently, DeepShadow [21] models the depth map of a scene by an MLP and optimizes the model with a shadow reconstruction loss. These methods can only recover a height map of the visible surface. Besides, they require the detected shadow regions as input, but shadow detection is itself a non-trivial problem.

**Neural scene representation**  Neural scene representations have been successfully applied in novel-view synthesis and multi-view reconstruction [38, 48, 52, 59, 63]. The popular neural radiance field (NeRF) [35] represents a continuous space with an MLP, which regresses the volume density and RGB color of a 3D point from the point coordinates and view direction. Attracted by the photo-realistic rendering produced by NeRF, many follow-up works are introduced to improve the reconstructed surface quality [39, 55, 64], rendering speed [14, 30, 41], optimization speed [37, 50, 65], and robustness [1, 33, 68].

The above methods consider each 3D point as an emitter, making them not able to model the surface materials and lighting separately. Inverse rendering methods have been proposed to jointly recover shape, materials, and lightings in a casual capture setup [3, 4, 69, 71, 72]. NeRV [49] explicitly models shadow and indirect illumination assuming a known environment map. NRF [2] and IRON [70] adopt a co-located camera-light setup to simplify the image formation model. PS-NeRF [62] utilizes multi-view and multi-light images to induce regularizations for more accurate surface reconstruction.

There are some attempts to reconstruct a radiance field from a single-view image in a data-driven manner (e.g., conditioning the MLP input with image features [13, 42, 66]), or utilizing depth image as shape prior [60]. However, due to the strong ambiguity, these methods struggle to achieve high-quality reconstruction. Compared with the above approaches, our method extends neural scene representation to reconstruct accurate shape and materials from single-view photometric stereo images.

# 3  Method

Given $N$ images captured from a single viewpoint under different near point lights, our method targets at recovering the geometry and materials for the scene (see Fig. 2). Following existing near-field photometric stereo methods [34, 44], we assume a calibrated perspective camera and known point light positions. Instead of representing the visible surface with a normal / depth map like others [25, 34, 44], we adopt a 3D neural field representation [3, 35, 39] to describe the 3D scene.

## 3.1  Neural Reflectance Field Representation

Our method is built on top of the recent neural radiance field (NeRF) [35]. Following UNISURF [39], we adopt an occupancy field instead of a density field to better represent the surface geometry. UNISURF uses an MLP to map a 3D point $\boldsymbol{x} \in \mathbb{R}^3$ and a view direction $\boldsymbol{d} \in \mathbb{R}^3$ to occupancy $o(\boldsymbol{x}) \in \mathbb{R}$ and color $c(\boldsymbol{x}, \boldsymbol{d}) \in \mathbb{R}^3$. An image can be generated through volume rendering in which the color of each pixel (or ray $\boldsymbol{r}$) is calculated by

$$\boldsymbol{C}(\boldsymbol{r}) = \sum_{i=1}^{N_V} o(\boldsymbol{x}_i) \prod_{j<i} \left(1 - o\left(\boldsymbol{x}_j\right)\right) c(\boldsymbol{x}_i, \boldsymbol{d}), \tag{1}$$

where $\boldsymbol{x}_i$ denotes a 3D point sampled along the ray $\boldsymbol{r} = \boldsymbol{o} + t\boldsymbol{d}$, with $\boldsymbol{o} \in \mathbb{R}^3$ being the camera center and $\boldsymbol{d} \in \mathbb{R}^3$ the ray direction specified by the pixel, and $N_V \in \mathbb{R}$ is the number of samples per ray.

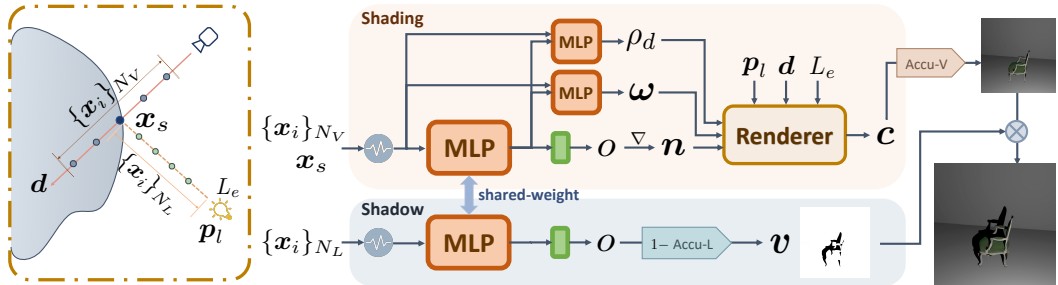

Figure 2: Overview of the method. For each camera ray, we first apply root-finding to locate the surface intersection point $x_s$. $N_V$ points on the camera ray are sampled within a relatively large interval around the surface to generate accumulated shading values. $N_L$ points are sampled on the surface-to-light segment to calculate the light visibility, which is multiplied to the accumulated shading to output the final RGB value.

Given multi-view images, a radiance field can be optimized to reproduce the input images. However, applying NeRF-based methods to single-view images is non-trivial. A straightforward idea would be to condition the color MLP of NeRF also on light directions, but our experiments show that such a naïve solution fails to produce reasonable reconstruction due to the lack of constraints on scene geometry.

To utilize shading information in photometric stereo images, we explicitly model the BRDFs of the scene and recover 3D point color with physics-based rendering [2, 3]. Observing that shadow provides strong cues for inferring the geometry of both visible and invisible surfaces in a scene, we compute shadow in an online manner by tracing a ray from a surface point to the light position to determine its light visibility. Give a point light located at $p_l \in \mathbb{R}^3$ with emitted intensity $L_e \in \mathbb{R}$, Eq. (1) can be rewritten as

$$C(r) = \sum_{i=1}^{N_V} o(x_i) \prod_{j<i} (1 - o(x_j)) f_v(p_l; x_i) f_c(d, p_l, L_e; x_i), \qquad (2)$$

where the 3D point color $c(x, d)$ is replaced by the product of light visibility $f_v(p_l; x)$ and physics-based rendered color $f_c(d, p_l, L_e; x)$, the details of which are given in the following subsections.

## 3.2  Physics-based Color Rendering

We consider non-Lambertian surfaces with spatially-varying BRDFs. The rendering equation for a surface point $x$ viewed from a direction $d$ under a near point light $(p_l, L_e)$ can be written as

$$f_c(d, p_l, L_e; x) = \underbrace{L_{int}(p_l, L_e; x)}_{\text{Light Intensity}} \underbrace{f_m(d, w_i(p_l; x); x)}_{\text{BRDF Value}} \underbrace{\max(w_i(p_l; x) \cdot n(x), 0)}_{\text{Shading}}, \qquad (3)$$

where $L_{int}(p_l, L_e; x)$ denotes the incident light (taking light falloff into account), $w_i(p_l; x)$ the incident light direction, and $f_m(d, w_i(p_l; x); x)$ the BRDF value at $x$. The normal at $x$ can be derived from the gradient of the occupancy field as $n(x) = \nabla o(x)/\|\nabla o(x)\|_2$ [39].

**Lighting model**  Following previous works [34, 44], we adopt the inverse-square law for point light attenuation where light intensity $L_{int}$ is proportional to the multiplicative inverse of the square of the distance $s$ (i.e., $L_{int} \propto 1/s^2$). The incident light direction $w_i$ and light intensity $L_{int}$ at a point $x$ are given by

$$w_i(p_l; x) = \frac{p_l - x}{\|p_l - x\|_2}, \qquad L_{int}(p_l, L_e; x) = \frac{L_e}{\|p_l - x\|_2^2}. \qquad (4)$$

**BRDF model**  Similar to [3, 69, 71], we adopt a BRDF model represented by a combination of diffuse color $\rho_d$ and specular reflectance $\rho_s$, which is given by

$$f_m(w_i, w_o; x) = \rho_d + \rho_s(w_i, w_o; x). \qquad (5)$$

Following [16, 25], we model the isotropic specular reflectance by a weighted combination of Sphere Gaussian (SG) bases, which demonstrates better results in modeling specular effects than the parametric Microfacet model [20]. The specular component $\rho_s$ is hence written as $\rho_s = \boldsymbol{\omega}^T D(\boldsymbol{h}, \boldsymbol{n})$, where $D(\boldsymbol{h}, \boldsymbol{n}) = G(\boldsymbol{h}, \boldsymbol{n}; \lambda) = \left[e^{\lambda_1(\boldsymbol{h}^T\boldsymbol{n}-1)}, \cdots, e^{\lambda_k(\boldsymbol{h}^T\boldsymbol{n}-1)}\right]^T$ denotes the SG bases, with $\lambda_* \in \mathbb{R}_+$ controls the specular sharpness. The diffuse component $\rho_d$ and SG weights $\boldsymbol{\omega}$ are estimated by two MLPs.

### 3.3 Online Shadow Computation

A 3D point $\boldsymbol{x}$ is shadowed if there is any occluders in its line of sight for the light position $\boldsymbol{p}_l$. It follows that light visibility $f_v(\boldsymbol{p}_l, \boldsymbol{x}) \in [0, 1]$ for a 3D point $\boldsymbol{x}$ can be computed by accumulating occupancies along this line (see Fig. 2), $i.e.$,

$$f_v(\boldsymbol{p}_l; \boldsymbol{x}) = 1 - \sum_{i=1}^{N_L} o(\boldsymbol{x}_i) \prod_{j<i} \left(1 - o\left(\boldsymbol{x}_j\right)\right), \tag{6}$$

where $N_L$ is the number of points sampled along the line.

However, calculating light visibilities for all $N_V$ points sampled along the ray for a pixel is computationally expensive ($i.e.$, $O(N_V N_L)$ MLP queries for each pixel / ray (see Fig. 3 (a)). To speed up shadow computation, previous methods either adopt an MLP to directly regress light visibility of a point [49] to reduce the queries for each ray to $O(N_V)$ (see Fig. 3 (b)), or pre-extracts the surface points (assuming a fixed scene geometry) [71] to reduce the number of MLP queries to $O(N_L)$. Instead, we first locate the expected surface points $\boldsymbol{x}_s$ along the ray by root-finding [39] and calculate its light visibility in an online manner. Eq. (2) can be reformulated for efficient color rendering as

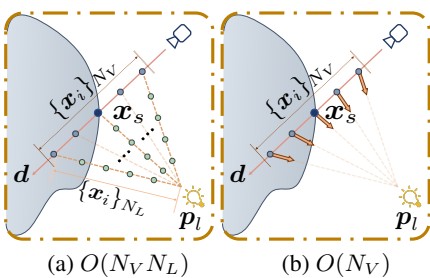

(a) $O(N_V N_L)$      (b) $O(N_V)$

Figure 3: Alternative shadow modeling.

$$\boldsymbol{C}(\boldsymbol{r}) = f_v(\boldsymbol{p}_l; \boldsymbol{x}_s) \sum_{i=1}^{N_V} o(\boldsymbol{x}_i) \prod_{j<i} \left(1 - o\left(\boldsymbol{x}_j\right)\right) f_c(\boldsymbol{x}_i, \boldsymbol{d}, \boldsymbol{p}_l, L_e). \tag{7}$$

### 3.4 Optimization

Different from shape-from-shadow methods [21, 53], our method does not require direct supervision for shadow rendering. We rely on image reconstruction loss for optimization.

**Volume rendering loss** The first loss is the L1 reconstruction loss between the volume rendered image $\boldsymbol{C}_v$ ($i.e.$, the computed $\boldsymbol{C}(\boldsymbol{r})$ in Eq. (7)) and the input image:

$$\mathcal{L}_v = \sum \|\boldsymbol{C}_v - I\|_1. \tag{8}$$

**Surface rendering loss** UNISURF [39] proposes to combine the volume rendering and surface rendering by gradually shortening the sampling range in a ray to refine the surface region. However, we empirically found that the model will start to degrade when the sampling interval is decreased as there is no multi-view information to constrain the non-sampled regions. We therefore propose to adopt a joint volume and surface rendering strategy. We additionally compute the surface rendering color $\boldsymbol{C}_s(\boldsymbol{r})$ using the expected surface point $\boldsymbol{x}_s$ and calculate the L1 loss, $i.e.$,

$$\mathcal{L}_s = \sum \|\boldsymbol{C}_s - I\|_1, \tag{9}$$

$$\boldsymbol{C}_s(\boldsymbol{r}) = f_v(\boldsymbol{p}_l; \boldsymbol{x}_s) f_c(\boldsymbol{d}, \boldsymbol{p}_l, L_e; \boldsymbol{x}_s). \tag{10}$$

**Normal smoothness loss** Similar to [39], we also include a regularization loss to promote smoothness in surface normal ($\epsilon$ is a small random perturbation):

$$\mathcal{L}_n = \sum \|\boldsymbol{n}(\boldsymbol{x}_s) - \boldsymbol{n}(\boldsymbol{x}_s + \epsilon)\|_2^2. \tag{11}$$

Table 1: Comparison with neural field methods on relighting and normal estimation results.

| Method | BUDDHA PSNR↑ | BUDDHA MAE↓ | READING PSNR↑ | READING MAE↓ | BUNNY PSNR↑ | BUNNY MAE↓ | CHAIR PSNR↑ | CHAIR MAE↓ | LEGO PSNR↑ | LEGO MAE↓ | HOTDOG PSNR↑ | HOTDOG MAE↓ |
|---|---|---|---|---|---|---|---|---|---|---|---|---|
| NeRF* [35] | 38.57 | 70.12 | 39.50 | 72.60 | 37.41 | 68.35 | 35.25 | 88.46 | **35.56** | 91.09 | **39.80** | 72.07 |
| UNISURF* [39] | 41.51 | 54.86 | 40.54 | 60.59 | 38.48 | 54.27 | 34.98 | 47.79 | 34.55 | 45.81 | 38.64 | 51.00 |
| Ours | **43.42** | **2.44** | **43.13** | **2.03** | **40.43** | **1.72** | **36.33** | **1.83** | 35.54 | **6.49** | 38.01 | **2.50** |

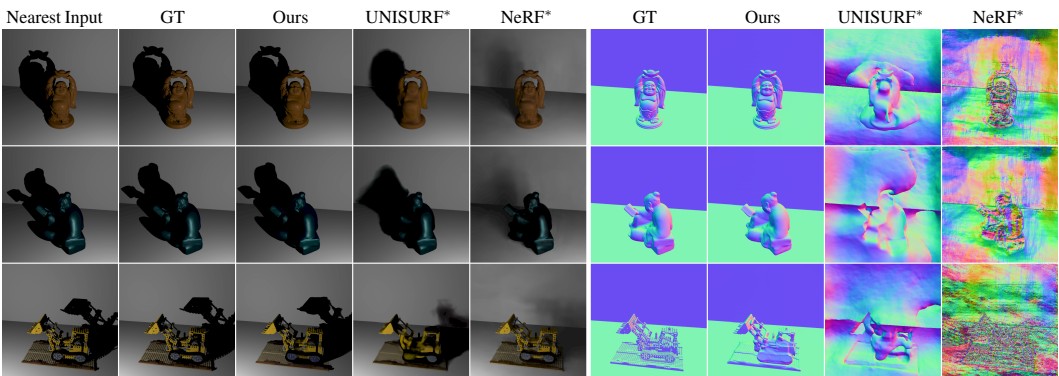

Figure 4: Comparison with neural field methods on relighting (left) and normal estimation (right).

**Overall loss**  The overall loss function used for optimization is as follow with $\alpha$ set to 0.005:

$$\mathcal{L} = \mathcal{L}_v + \mathcal{L}_s + \alpha\mathcal{L}_n. \tag{12}$$

# 4 Experiments

## 4.1 Implementation Details

Similar to UNISURF [39], we use an 8-layer MLP (256 channels with softplus activation) to predict the occupancy $o$ and output a 256-dimensional feature vector. Two additional 4-layer MLPs then take the feature vector and point coordinates as input to predict the albedo $\rho_d$ and weights $\omega$ of SG bases. We sample $N_V = 256$ points along the camera ray and $N_L = 256$ points along the surface-to-light line segment. We use Adam optimizer [23] with an initial learning rate of 0.0002 which decays at 200 and 400 epochs. We train each scene for 800 epochs on one Nvidia RTX 3090 card, which takes about 16 hours to converge.

**Evaluation Metrics**  We adopt mean angular error (MAE) in degree for surface normal evaluation and L1 error in $cm$ for depth assessment. PSNR is used to measure the quality of images rendered under novel view or novel lighting.

## 4.2 Datasets

Our method targets at recovering the complete scene by exploiting both shading and shadow. However, existing photometric stereo datasets are mostly interested in the object region and intentionally remove the influence of the background (*e.g.*, cover the background with black cloth to avoid inter-reflections [47]), which makes the shadow and shading information invisible in the background regions. Therefore, such datasets [34, 47] are not suitable to evaluate the full potential of our method.

Instead, we evaluate our method on multiple synthetic datasets with complicated scene geometry and materials. Specifically, we used 10 3D objects for data rendering, where 5 objects from DiLiGent-MV Dataset [27] (namely, *BEAR*, *BUDDHA*, *COW*, *POT2*, and *READING*), 2 objects from the internet (namely, *BUNNY* and *ARMADILLO*), and 3 objects from NeRF's blender dataset [35] (namely, *LEGO*, *CHAIR*, and *HOTDOG*). We rendered *LEGO*, *CHAIR*, and *HOTDOG* with Blender's Cycles pathtracer, and the other 7 objects with Mitsuba [19]. As our method does not explicitly model inter-reflections, we set the max bounces to 0 during rendering. During rendering, we created a scene by adding a horizontal and a vertical plane to model the desk and wall, and objects are placed on the

Table 2: Comparison with single-view normal / depth estimation methods (only object regions).

| Method | BUDDHA | | READING | | BUNNY | | CHAIR | | LEGO | | HOTDOG | |
| | MAE↓ | Depth L1↓ | MAE↓ | Depth L1↓ | MAE↓ | Depth L1↓ | MAE↓ | Depth L1↓ | MAE↓ | Depth L1↓ | MAE↓ | Depth L1↓ |
|---|---|---|---|---|---|---|---|---|---|---|---|---|
| ZL18 [28] | 37.51 | 19.84 | 37.29 | 25.97 | 31.40 | 17.68 | 39.53 | 41.19 | 46.82 | 34.56 | 39.74 | 18.02 |
| QY18 [40] | **12.25** | 3.81 | 40.84 | 26.13 | 14.21 | 4.10 | 29.68 | 15.95 | 33.08 | 17.87 | 16.81 | 8.98 |
| HS20 [44] | 18.39 | 6.47 | 27.11 | 18.94 | 16.92 | 10.96 | 29.56 | 13.99 | 33.54 | 13.27 | 27.25 | 13.22 |
| Ours | 14.24 | **1.50** | **7.00** | **2.09** | **9.40** | **1.63** | **17.43** | **4.74** | **31.13** | **7.31** | **14.65** | **1.68** |

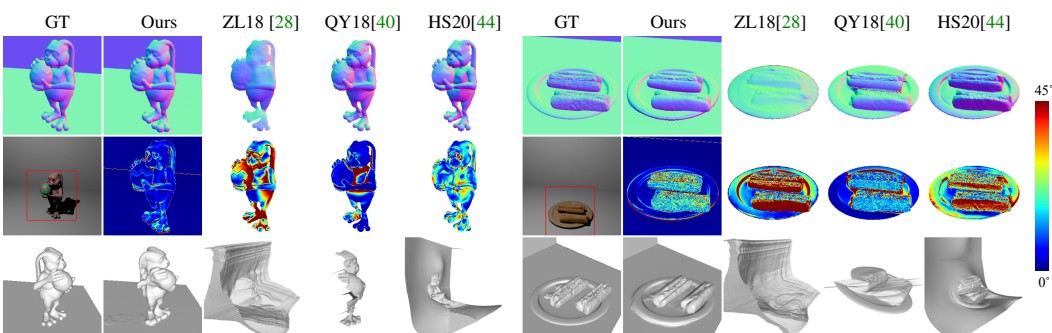

Figure 5: Comparison with single-view normal / depth estimation baselines. Row 1 and Row 2 show the normal and error maps. Row 3 shows the side-view of the reconstructed surfaces.

horizontal plane. Each scene was rendered under $128$ uniformly sampled near point lights, and the rendered images are in linear space with a resolution of $512 \times 512$.

## 4.3 Comparisons with Existing Methods

To justify the effectiveness of our method, we compare it with three types of methods, namely, neural field methods, photometric stereo methods, and single-image shape estimation methods.

**Neural radiance field methods** We first verify the design of our method by comparing it with two simple baselines (*i.e.*, adapting NeRF [35] and UNISURF [39] for this problem by conditioning the color MLP on light direction). Table 1 and Fig. 4 show the normal estimation and relighting results in the training view. Although the baseline methods can achieve reasonable rendering results in terms of PSNR, they fail to predict accurate cast-shadow and cannot reconstruct the geometry of the scene (with a large average MAE of 77.12/52.39). In contrast, our method is able to accurately reconstruct the shape with an average MAE of 2.84, and achieves the best rendering results (average PSNR of 39.48). This result indicates that simply conditioning the color MLP on light direction does not provide sufficient constraint to regularize the scene geometry.

**Single-view shape estimation methods** We then compare with three state-of-the-art single-view normal / depth estimation methods, including two near-field PS methods (QY18 [40] and HS20 [44]) and one single-image shape estimation method (ZL18 [28]). QY18 [40] and HS20 [44] consider exactly the same setup as our method (multiple images captured under near point lights), so the input are the same as our method. ZL18 [28] assumes an image captured under co-located flash light as input, so we choose the image illuminated by a point light that is closest to the camera as its input. As these methods are designed to estimate the shape in the object region and have difficulty in dealing with the background, we only report the normal and depth estimation results on the object region for the training view in Table 2. Since ZL18 [28] and HS20 [44] require depth alignment before evaluation, we align the estimated depth with the ground truth for all the methods for fair comparison. We can see that our method achieves the best results for both normal and depth estimation. Moreover, as shown in Fig. 5, our method can faithfully reconstruct both visible and invisible parts of the scene, which is not possible by methods that rely on the normal or depth representation.

## 4.4 Method Analysis

We next conduct ablation study for different components of our method, and evaluate our method on different setups to further analyze its behavior.

Table 3: Quantitative results for the ablation study.

| | Train View | | | | | | Novel Views | | | | | |
| | CHAIR | | BUNNY | | BUDDHA | | CHAIR | | BUNNY | | BUDDHA | |
| Method | MAE↓ | PSNR↑ | MAE↓ | PSNR↑ | MAE↓ | PSNR↑ | MAE↓ | PSNR↑ | MAE↓ | PSNR↑ | MAE↓ | PSNR↑ |
|---|---|---|---|---|---|---|---|---|---|---|---|---|
| w/o shading | 32.49 | 33.71 | 40.18 | 38.72 | 35.68 | 41.43 | – | – | – | – | – | – |
| w/o shadow | 3.39 | 30.81 | 2.26 | 33.45 | 3.33 | 34.30 | 12.24 | 22.31 | 11.93 | 24.43 | 16.60 | 23.27 |
| w/o $\mathcal{L}_s$ | 2.48 | 35.85 | 2.75 | 39.73 | 3.77 | 43.04 | **5.10** | **28.58** | 6.27 | 29.11 | 8.50 | 28.61 |
| Ours | **1.83** | **36.33** | **1.72** | **40.43** | **2.44** | **43.42** | 5.45 | 26.82 | **6.11** | **29.55** | **6.89** | **31.53** |

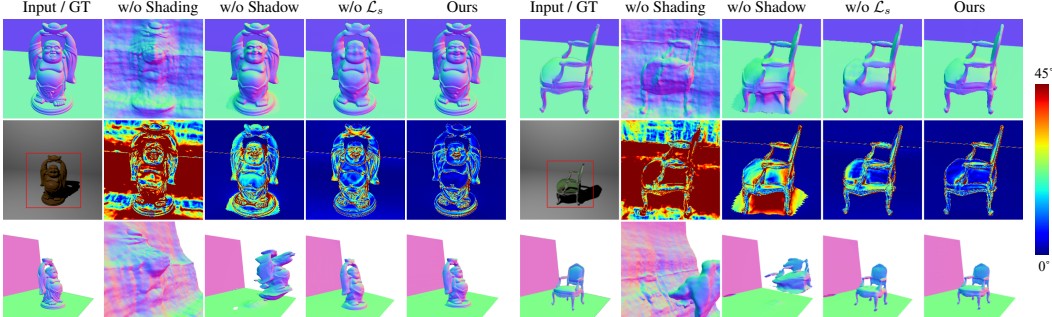

Figure 6: Visual results for the ablation study. Row 1 is the normal of train view, and row 2 shows its error map compared with ground truth. Row 3 shows the normal of a novel view.

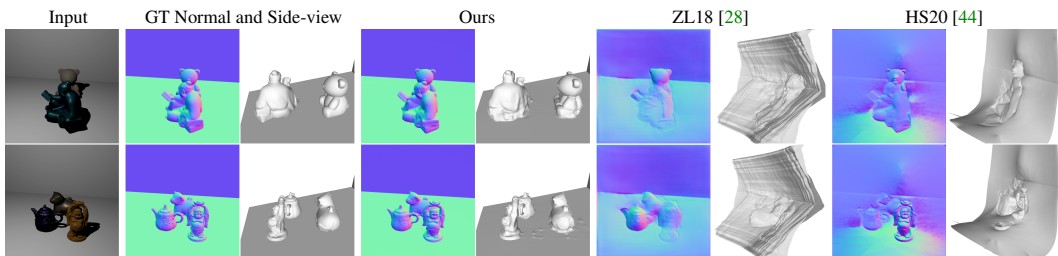

Figure 7: Results on scenes with multiple occluding objects.

**Joint shading and shadow modeling** Our method exploits both shading and shadow information for scene reconstruction. To analyze the effect of both components in reconstructing the scene geometry, we trained two variant models where (a) *"w/o shading"* replaces the BRDF module with a 4-layer MLP to directly predict RGB values with additional light location input; and (b) *"w/o shadow"* removes the shadow module and only output the shading. Results are summarized in Table 3 and Fig. 6. We evaluate the results for both trained view and novel views, and report MAE of normal maps and PSNR of rendered images. As the *"w/o shading"* model fails to estimate proper depth and the recovered surfaces totally deviate from the ground truth, we omit its results for novel views. Without shadow information, the model may still predict proper surface normal for the trained view. However, the model fails to predict the depth and shape of the object since there is no constraint on invisible regions. By exploiting both shading and shadow cues, our method can well reconstruct the full scene.

**Joint volume and surface rendering** We also analyze the effectiveness of the surface rendering loss $\mathcal{L}_s$ by comparing our full model with the one without $\mathcal{L}_s$. Results in Table 3 and Fig. 6 show that surface rendering loss can effectively refine the surface normals of the object surface.

**Effect of occlusion/discontinuity and unseen region** We further demonstrate the potential of our method in reconstructing the complete geometry of the scene, especially when there are occlusions or discontinuous surfaces. Figure 7 shows the reconstructions for two scenes with multiple objects occluding each other. It is very difficult to identify the shape of the invisible regions just from the single-view images. However, by effectively leveraging the shadow information, our method successfully predicts the shape (*i.e.*, the occupancy field) of the invisible regions, which is not feasible for existing works.

We also investigate the performance of our method on surface with challenging invisible shapes. Note that the shape of invisible regions are mainly constrained by the shadow (which indicates the

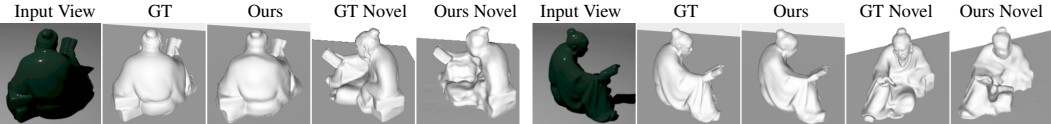

| Input View | GT | Ours | GT Novel | Ours Novel | Input View | GT | Ours | GT Novel | Ours Novel |

Figure 8: Results on two scenes with challenging invisible shapes.

Table 4: Analysis on Light numbers.

| | CHAIR | | | ARMADILLO | | |
|---|---|---|---|---|---|---|
| Light# | MAE↓ | Depth↓ | PSNR↑ | MAE↓ | Depth↓ | PSNR↑ |
| 4 | 30.39 | 181.62 | 19.39 | 46.87 | 93.66 | 15.87 |
| 8 | 2.33 | 10.99 | 35.60 | 2.51 | 5.98 | 37.17 |
| 16 | 2.10 | **8.11** | 36.20 | 2.38 | **5.89** | 37.50 |
| 32 | 1.97 | 8.77 | 36.23 | 2.05 | 6.27 | 39.20 |
| 64 | **1.81** | 8.64 | **36.52** | 2.00 | 7.18 | 39.78 |
| 128 | 1.83 | 9.04 | 36.33 | **1.88** | 6.65 | **40.13** |

Table 5: Analysis on Light Range.

| | CHAIR | | | ARMADILLO | | |
|---|---|---|---|---|---|---|
| Range | MAE↓ | Depth↓ | PSNR↑ | MAE↓ | Depth↓ | PSNR↑ |
| small | 3.92 | 18.79 | 30.15 | 2.32 | 6.86 | 35.26 |
| median | 1.93 | **8.75** | 35.96 | **1.70** | **4.59** | 38.92 |
| broad | **1.83** | 9.04 | **36.33** | 1.88 | 6.65 | **40.13** |

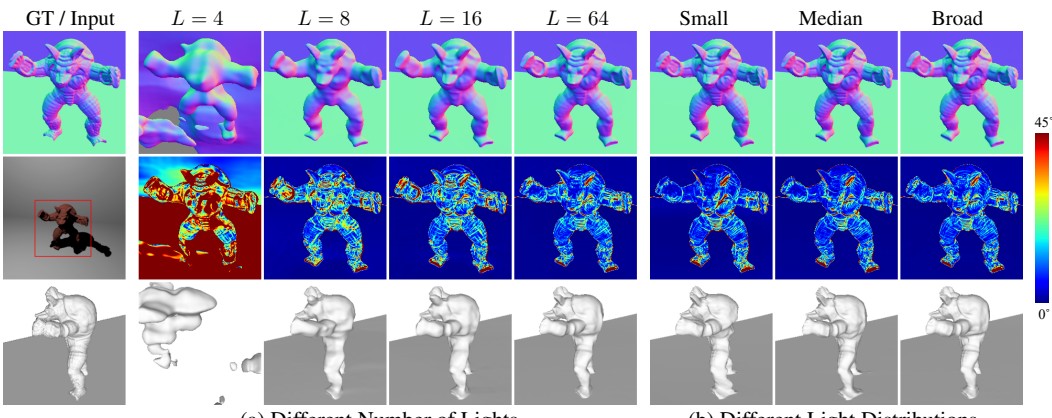

| GT / Input | $L = 4$ | $L = 8$ | $L = 16$ | $L = 64$ | Small | Median | Broad |

(a) Different Number of Lights        (b) Different Light Distributions

Figure 9: Analysis on different number of lights and light distributions.

occupancy along the light path). In Fig. 8, we show the reconstruction of *READING* object which is posed to make the concave surface invisible. From the results of novel views, we can see that our method can properly recover the invisible irregular surface, though some invisible regions are not fully consistent with the ground truth shape. This result demonstrates that shadow provides strong cue for shape recovery especially for unseen regions.

**Effect of light distributions** To analyze the robustness of our method on different light distribution, we evaluate it on scenes illuminated by different number of lights or different ranges (see our supplementary material for visualization of light distributions), and the numerical results are summarized in Table 4 and Table 5 (depth errors are calculated in object regions). The model fails to reconstruct the scene with only 4 light inputs, but can reconstruct faithful shape of the scene with 8 light inputs. With more lights used, the surface of the object is further refined (see Fig. 9). We can also observe that our method can still work for small range of light distributions to recover invisible regions. Overall, our method is robust to different number and different range of lights.

**Results on real scenes** To further demonstrate the practicality of our method, we evaluate on three real scenes, which were captured using a fixed camera (with 28mm focal length) and a handheld cellphone flashlight (see Fig. 10). The object was put on the table and close to the wall. We turned off all the environmental light sources and only kept the flashlight on, which was randomly moved around to capture images illuminated under different light conditions. For each object we took around 70 images.

Our setup does not require manual calibration of lights. Instead, we applied the state-of-the-art self-calibrated photometric stereo network (SDPS-Net [8]) for light direction initialization, and roughly measured the camera-object distance as initialization of light-object distance. After initialization, the position and direction of lights are jointly optimized with the shape and BRDF during training. Please refer to our supplementary materials for more training details. Sample inputs and results are

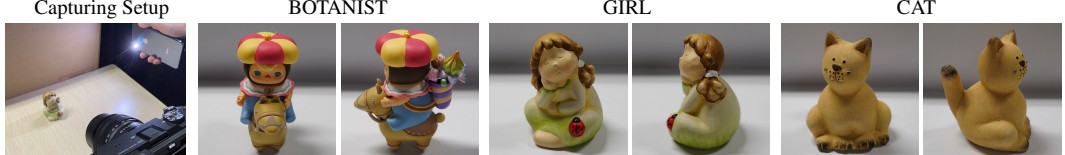

Capturing Setup    BOTANIST    GIRL    CAT

Figure 10: The data capturing setup and three testing objects.

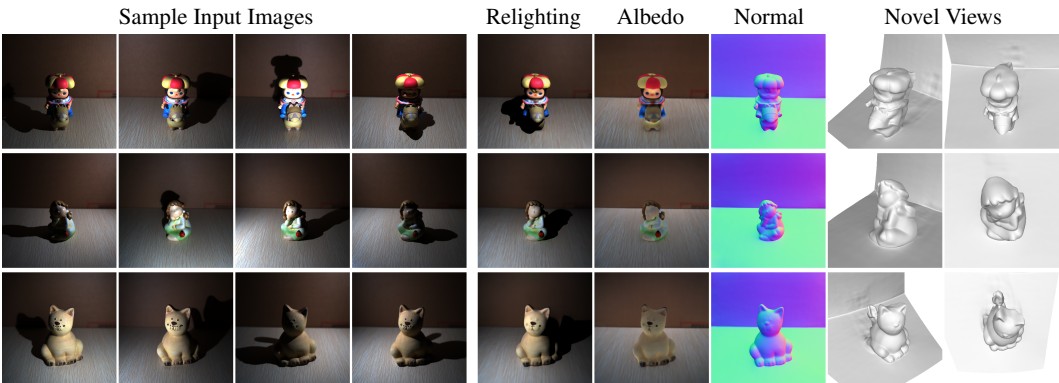

Sample Input Images    Relighting    Albedo    Normal    Novel Views

Figure 11: Results on the real captured data. From top to bottom: *BOTANIST*, *GIRL*, *CAT*.

shown in Fig. 11. Even with this casual capturing setup and uncalibrated lights, our method achieved satisfactory results in normal prediction and full 3D shape reconstruction.

## 5 Conclusion

In this paper, we have introduced a method to optimize a neural reflectance field for a non-Lambertian scene from single-view images captured under different near point lights. Our method jointly recovers the geometry (*i.e.*, occupancy field) and BRDFs of the scene by fully utilizing the shading and shadow cues. Interestingly, our results on scenes with complicated shapes and materials show that the complete scene geometry can be faithfully reconstructed just from single-view photometric images. Moreover, comprehensive method analysis demonstrates that our method is robust to scenes with different geometry, materials, light number, and light distributions. Additionally, our method supports applications like novel-view synthesis and relighting.

**Limitation**   First, like existing near-field PS methods [44], our method requires known light positions, which requires additional efforts for lighting calibration. Second, as our method relies on shadow cue for invisible shape reconstruction, its performance may decrease if the scene background is highly complicated as the background geometry will affect the appearance of shadows. Third, although the shape of the invisible parts can be well reconstructed, the reflectance of those regions are not well constrained by shadow. Last, our method ignores inter-reflection effects in image formation. In the future, we will further extend our method to tackle these limitations.

**Acknowledgements**   This work was partially supported by the National Key R&D Program of China (No.2018YFB1800800), the Basic Research Project No. HZQB-KCZYZ-2021067 of Hetao Shenzhen-HK S&T Cooperation Zone, NSFC-62202409, and the Research Grant Council of Hong Kong (SAR), China (project no. 17203119).

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
