# $S^3$-NeRF: Neural Reflectance Field from Shading and Shadow under a Single Viewpoint

## —— Supplementary Materials ——

In this supplementary material, we provide more results, analyses, implementation details and discussions to supplement our main paper.

## Contents

# A Visual Examples For the Shadow Cue

To help better understand how shadow provides cues for inferring shape of the invisible surface, in Fig. S1, we visualize the rendered images of three different objects, which have the same front view but with different shapes in the back (generated by cutting the *READING* mesh with a plane). We can see that although these three objects have the same shapes and appearances in the front view, the produced shadows are largely different, demonstrating that shadow can provide strong information for shape reconstruction.

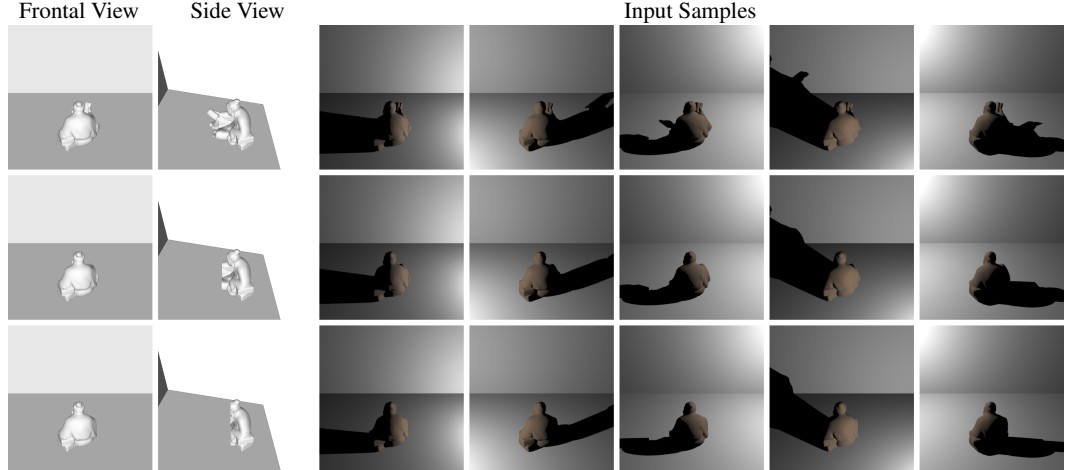

Figure S1: Visual examples to illustrate the shadow cue.

# B More Details for the Method

The detailed architecture of the network is visualized in Fig. S2. Similar to [6], we use *SoftPlus* activation for the occupancy branch and *ReLU* activation for albedo and specular weights branch. Following most neural rendering works, we adopt positional encoding (with hyper-parameter $L = 6$) to map the point coordinates to higher dimensions, which is then concatenated with the coordinate as the input. To stabilize the training process, we add the shadow modeling after 1K iterations, and the surface loss after 5K iterations.

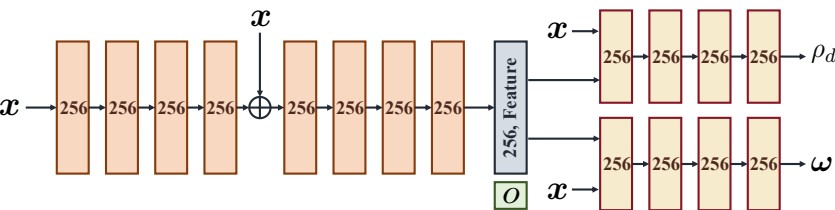

Figure S2: Detailed architecture of the network. Positional encoding is employed for the input $x$.

# C More Details for the Synthetic Dataset

We use both Mitsuba and Blender for rendering. Specifically, Blender is used for the *LEGO*, *CHAIR*, and *HOTDOG*, while other objects are rendered via Mitsuba. We created a scene by adding a horizontal and a vertical plane to model the desk and wall, and objects are placed on the horizontal plane. Each scene was rendered under $128$ uniformly sampled near point lights. We use the default materials for the Blender scenes and *BUNNY*, while employing the MERL dataset [4] to randomly select materials for the other 6 objects.

The light distribution used in the default experiment setups is shown as Fig. S3 (a). The small range and median range light distributions used in light range analysis (see Table 5 of the paper) are shown in Fig. S3 (b)-(c), respectively.

Figure S4 visualizes the light distributions used in light number analysis (see Table 4 of the paper).

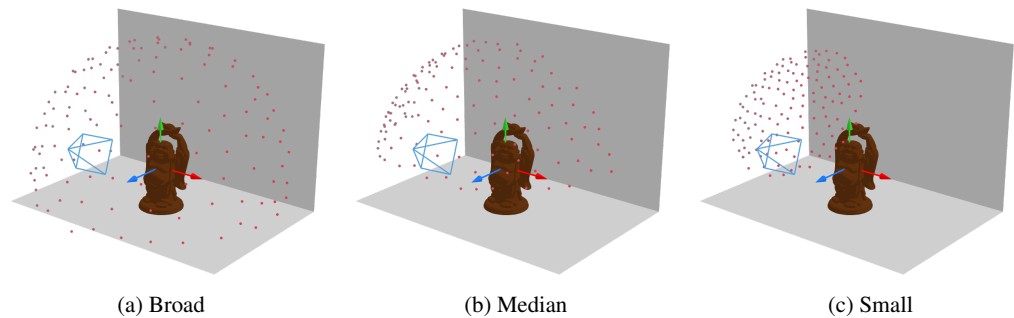

|  (a) Broad  |  (b) Median  |  (c) Small  |

Figure S3: Visualization of the light distributions with different ranges.



| (a) $L = 4$ | (b) $L = 8$ | (c) $L = 16$ | (d) $L = 32$ | (e) $L = 64$ |

Figure S4: Visualization of light distributions with different numbers of lights.

## D  More Method Analysis

### D.1  Results on Scenes with Different Backgrounds

To verify the capability of our method in dealing with scenes with different types of backgrounds, we evaluated it on four common types of backgrounds, namely the *Wall and Desk*, *Wall only*, *Desk only*, and *Wall Corner* (see Fig. S5). We can see that our method works well on different scene layouts, demonstrating the robustness of our method.

### D.2  Analysis on Complicated Background

To further evaluate the robustness of our method on more complicated backgrounds, we evaluated it on four scenes rendered with different backgrounds, including two uniform color backgrounds with different lightness (denoted as 'Light' and 'Dark') and two textured backgrounds. Results in Table S1 and Fig. S6 show that our method is robust to backgrounds with different lightness and textures.

Table S1: Results on background with different lightness and textures.

| BG Color | BUNNY | | READING | |
| --- | --- | --- | --- | --- |
| | MAE↓ | Depth↓ | MAE↓ | Depth↓ |
| White | 1.72 | 5.39 | 2.03 | 5.65 |
| Gray | 2.11 | 6.15 | 2.16 | 7.19 |
| Texture 1 | 1.93 | 8.30 | 2.36 | 8.75 |
| Texture 2 | 1.94 | 8.69 | 2.43 | 10.10 |

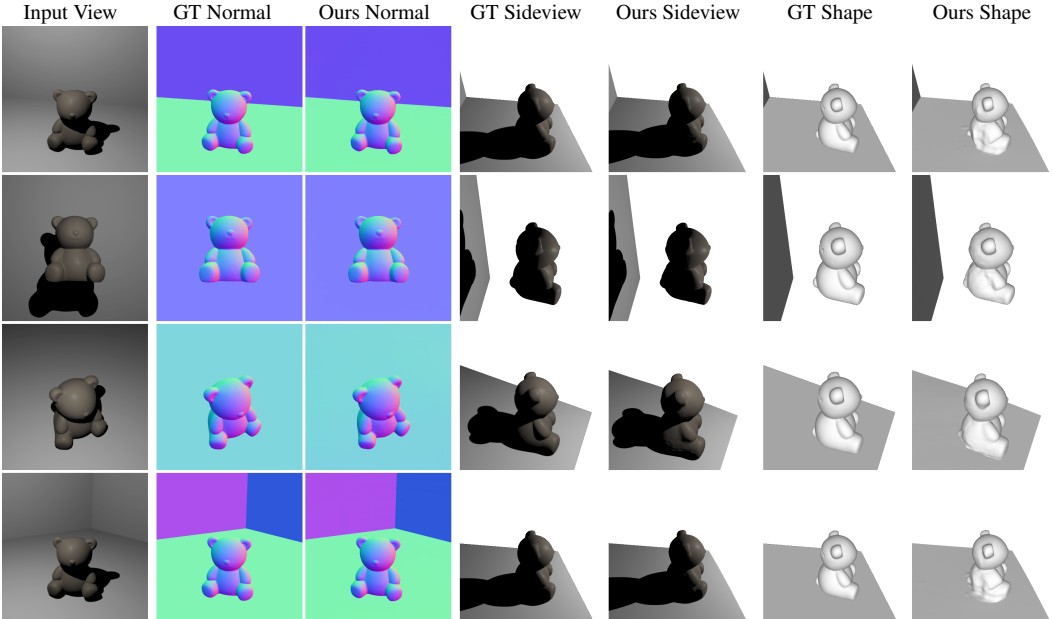

| Input View | GT Normal | Ours Normal | GT Sideview | Ours Sideview | GT Shape | Ours Shape |

Figure S5: Results on scenes with different background. From top to bottom shows the results on background with types of *Wall and Desk*, *Wall only*, *Desk only*, and *Wall Corner*.

### D.3    Analysis on Shadow Modeling in Foreground and Background Regions

We also analyze the effect of cast shadow modeling in both foreground and background regions. Specifically, we trained two variant models, one without foreground shadow modeling and the other without background modeling. Results in Table S2 and Fig. S7 show that modeling cast shadow in both regions is important, as disabling either one of them leads to decreased accuracy.

Table S2: Analysis of foreground/background shadow modeling (depth object regions only).

| Method | BUNNY | | CHAIR | |
| | MAE↓ | Depth↓ | MAE↓ | Depth↓ |
| --- | --- | --- | --- | --- |
| w/o back | 1.84 | 34.60 | 3.58 | 29.49 |
| w/o fore | 2.11 | **6.75** | 2.03 | 9.67 |
| Ours | **1.72** | 6.82 | **1.83** | **9.04** |

### D.4    Compare with MLP Regression for Shadow Computation

We also compare our shadow modeling method with direct MLP regression. We trained a variant model replacing the ray-marching visibility computation with a direct visibility MLP. Results in Table S3 and Fig. S8 show that simply regressing the visibility produces worse results, as this MLP cannot regularize the occupancy field. In contrast, our method performs ray-marching in the occupancy field to render shadow, providing strong constraints for the occupancy field.

Table S3: Comparison of our ray-marching shadow computation and MLP regression.

| Method | CHAIR | | | BUDDHA | | |
| | MAE↓ | Depth↓ | PSNR↑ | MAE↓ | Depth↓ | PSNR↑ |
| --- | --- | --- | --- | --- | --- | --- |
| Vis-MLP | 3.07 | 17.14 | 35.57 | 2.59 | 19.71 | 41.24 |
| Ours | **1.83** | **5.57** | **36.33** | **2.44** | **5.48** | **43.42** |

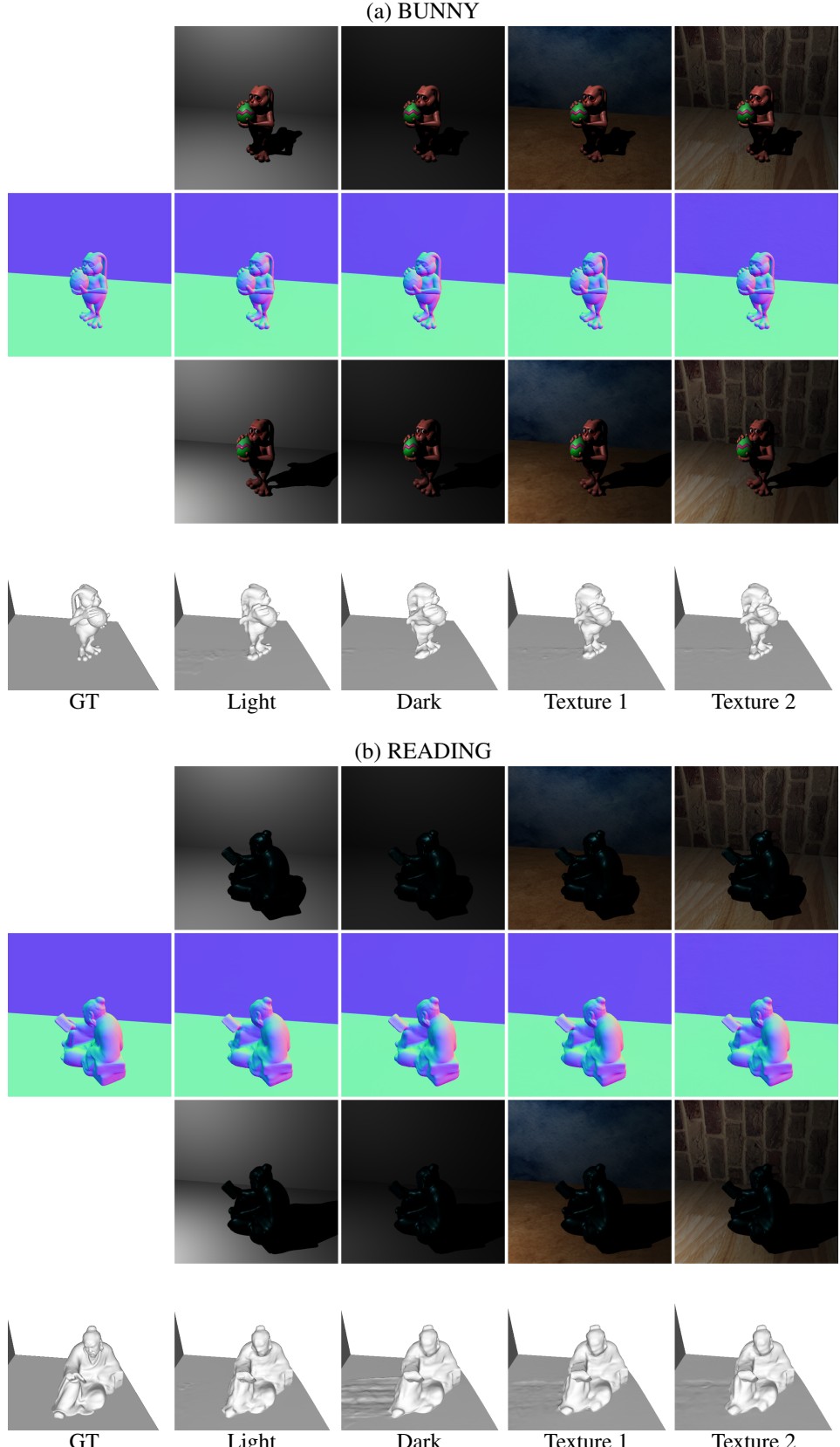

Figure S6: Visual results on backgrounds with different lightness and textures. Row 1 is the input sample, and row 2 shows the normal map of the view. Row 3 shows a rendered image under a novel light, and row 4 shows the shape of a novel view. (To make the lightness/texture details clearer, we show the GT/rendered images in linear space.)

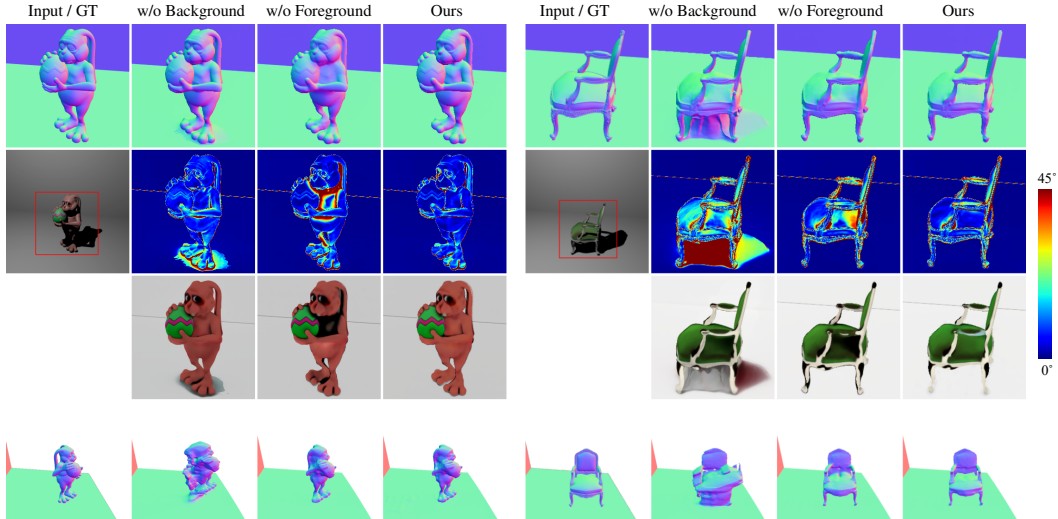

Figure S7: Visual results for the analysis on foreground/background shadow modeling. Row 1 is the normal of train view, and row 2 shows its error map compared with ground truth. Row 3 shows the albedo map and row 4 shows the normal of a novel view.

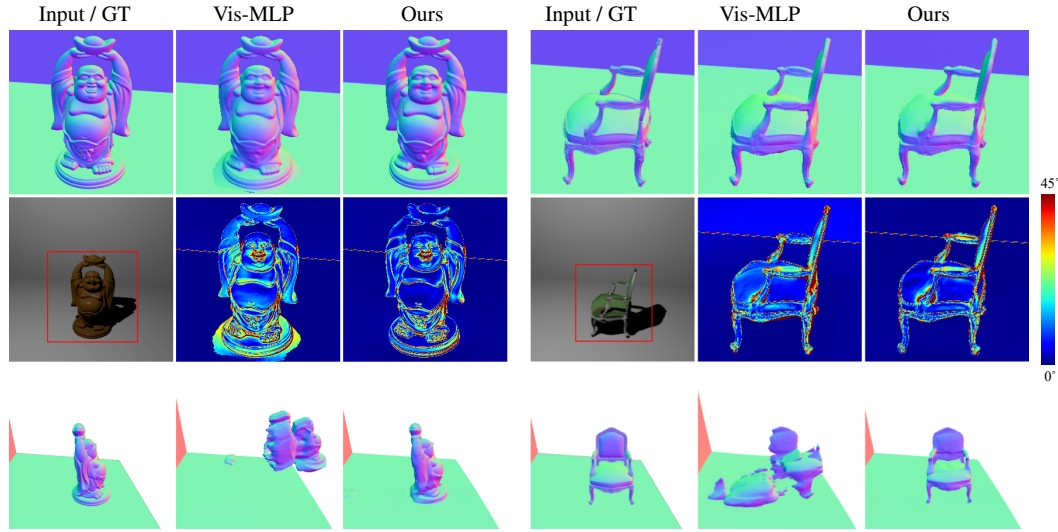

Figure S8: Visual results for the analysis on shadow modeling. "Vis-MLP" means using an MLP to predict the visibility distribution. Row 1 is the normal of train view, and row 2 shows its error map compared with ground truth. Row 3 shows the normal of a novel view.

## D.5 Effect of Area Light

We also analyze the effect of soft shadow caused by a larger light source, we tested our method on data rendered using light sources with different scales (i.e., a sphere with a radius of 1/50, 1/25, or 1/10 of the object size). Results in Fig. S9 show that our method is robust to larger light sources (e.g., 1/50 and 1/25). We also observe that when the light source size is considerably large (e.g., 1/10), the results in the object boundary will decrease because of the heavy soft shadow. Note that this is not a problem in practice as it is very easy to find a point light source whose size is smaller than 1/25 of the object size (e.g., the cellphone flashlight).

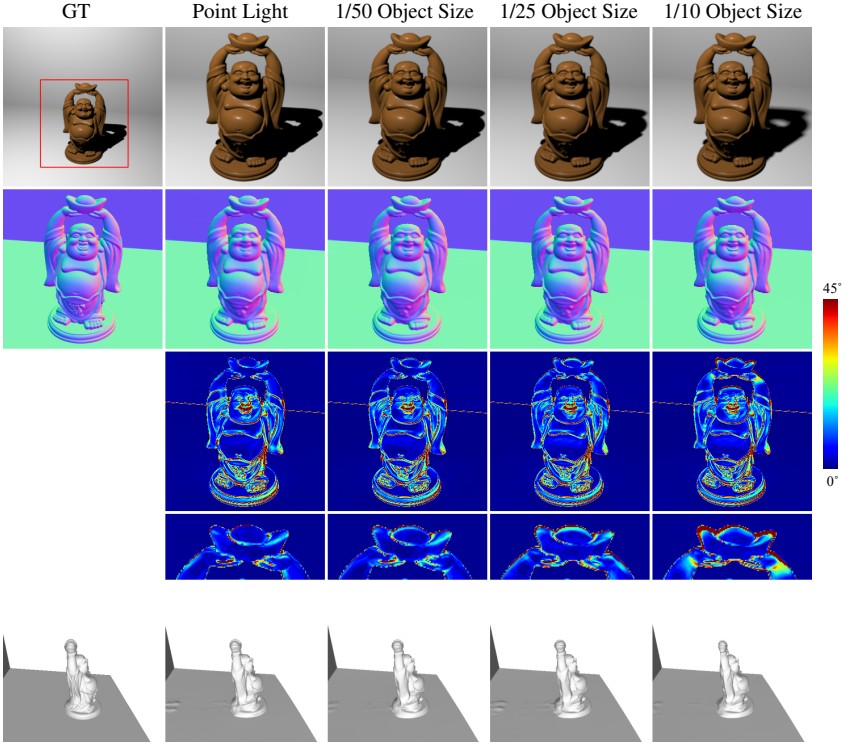

Figure S9: Visual results for the analysis of foreground/background shadow modeling. Row 1 is the input sample. Row 2 shows the normal map of the view, and row 3-4 shows its error map. Row 5 shows the surface in novel view.

## D.6 Effect of Lighting Distributions For Invisible Shape Reconstruction

To analyze the effect of light distribution on reconstructed shape of invisible regions, we also report the Chamfer Distance between the reconstructed and ground-truth meshes of "*ARMADILLO*" (object regions only), which can quantify the full shape reconstruction. Since the extracted scene consists of both the object and background, we crop out the background regions and only calculate the Chamfer Distance on objects. We also notice that the depth variance will cause significant increase of the errors. Therefore, we crop the bottom areas of the object and apply ICP to align the extracted mesh and the ground truth before calculating the Chamfer Distance. Results in Table S4 and Table S5 show that the shape accuracy will improve given more lights, and our method is able to achieve robust results given 8 input lights. When the light distribution becomes narrow (small), the shape accuracy will decrease.

## D.7 Effect of Normal Smoothness Loss

To further study the impact of the normal smoothness loss, we did an ablation study on the loss term. Results in Fig. S10 show that imposing the normal smoothness loss is helpful to reduce the artifacts in the invisible regions.

| | Table S4: Chamfer distance of model trained with different light numbers. | | Table S5: Chamfer distance of model trained with different light range. | |
|---|---|---|---|---|

**Table S4: Chamfer distance of model trained with different light numbers.**

| Light# | Chamfer Dist. ↓ |
|---|---|
| 4 | – |
| 8 | 10.16 |
| 16 | 8.08 |
| 32 | 7.42 |
| 64 | 7.74 |
| 128 | **6.92** |

**Table S5: Chamfer distance of model trained with different light range.**

| Range | Chamfer Dist. ↓ |
|---|---|
| small | 10.32 |
| median | **5.98** |
| broad | 6.92 |

| Input View | GT Novel View | w/o $\mathcal{L}_n$ | Ours |
|---|---|---|---|

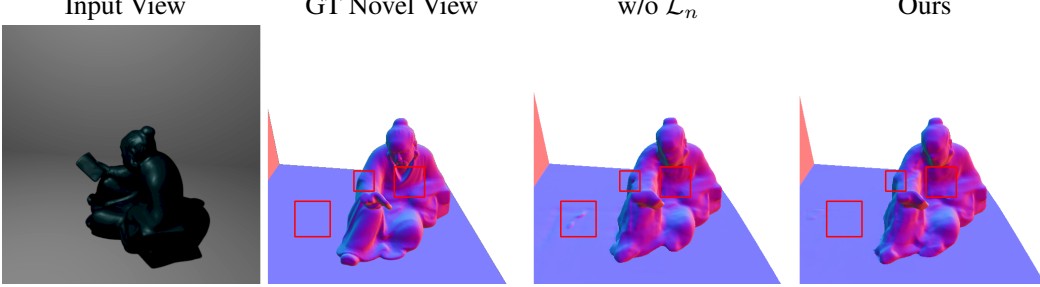

Figure S10: Ablation for normal smoothness loss.

# E  Results on the LUCES Dataset

As mentioned in Section 4.2 of the paper, existing photometric stereo (PS) datasets [5, 9] are primarily interested in the object region, and the shadow and shading information cannot be observed in the background regions. Therefore, they are not suitable to evaluate our method in *full* scene reconstruction.

For completeness of the evaluation, we compare our method with existing near-field PS methods on the public near-field PS dataset LUCES [5] for normal and depth estimations of the visible surface[1]. Note that only the ground-truth normal and depth maps of the object observed in the input view are provided. Following previous methods, we adopt an anisotropic light source [5] for light modeling.

As shown in Table S6 and Table S7, our method achieves the best average normal estimation result, and the depth estimation results (aligned) are comparable to state-of-the-art methods, even though this dataset does not well fit our assumption (*i.e.*, shading and shadow are observed in the background). Note that the results of other methods are collected from [5]. This result indicates that our method works well for real-world datasets with challenging geometry and materials, demonstrating the effectiveness of our method.

Table S6: Normal MAE of the input view on LUCES Dataset (object region only).

| Method | Bell | Ball | Buddha | Bunny | Die | Hippo | House | Cup | Owl | Jar | Queen | Squirrel | Bowl | Tool | Average |
|---|---|---|---|---|---|---|---|---|---|---|---|---|---|---|---|
| L17 [2] | 28.25 | 9.77 | 11.5 | 20.15 | 11.95 | 15.42 | 29.69 | 30.76 | 13.77 | 10.56 | 13.05 | 15.93 | 12.5 | 15.1 | 17.03 |
| I18 [1] | 23.55 | 44.29 | 35.29 | 36 | 41.52 | 44.9 | 49.05 | 35.78 | 40.27 | 40.66 | 32.89 | 41.09 | 28.04 | 31.71 | 37.5 |
| Q18 [7] | 25.8 | 12.12 | 14.07 | 13.73 | 13.77 | 18.51 | 30.63 | 37.63 | 14.74 | 15.66 | 13.16 | 14.06 | 11.19 | 16.12 | 17.94 |
| S20 [8] | 9.5 | 25.42 | 19.17 | 12.5 | 5.23 | 23.12 | 28.02 | **14.22** | 13.08 | 9.27 | 16.62 | 14.07 | 12.44 | 17.42 | 15.72 |
| L20 [3] | 14.74 | 12.43 | **10.73** | 8.15 | 6.55 | 7.75 | 30.03 | 23.35 | 12.39 | 8.6 | **10.96** | 15.12 | 8.78 | 17.05 | 13.33 |
| Ours | **7.66** | **5.96** | 12.67 | **7.38** | **3.67** | **6.26** | 27.61 | 30.19 | **8.78** | **5.49** | 11.37 | **12.45** | **6.11** | **12.25** | **11.28** |

Table S7: Depth L1 error of the input view on LUCES Dataset (object region only).

| Method | Bell | Ball | Buddha | Bunny | Die | Hippo | House | Cup | Owl | Jar | Queen | Squirrel | Bowl | Tool | Average |
|---|---|---|---|---|---|---|---|---|---|---|---|---|---|---|---|
| L17 [2] | 4.45 | 0.81 | 4.67 | 7.51 | 4.58 | 3.19 | 6.99 | 2.67 | 3.64 | 6.56 | **1.89** | 1.82 | 4.37 | 3.25 | 4.02 |
| I18 [1] | 5.93 | 6.59 | 10.92 | 6.88 | 7.83 | 7.59 | 8.98 | 3.17 | 8.67 | 15.54 | 8.08 | 5.8 | 6.69 | 12.45 | 8.22 |
| Q18 [7] | 12.03 | 2.5 | 9.28 | 7.06 | 5.91 | 6.8 | 8.02 | 4.83 | 5.83 | 16.87 | 6.92 | 2.55 | 6.48 | 6.69 | 7.27 |
| S20 [8] | 1.9 | 5.5 | 5.53 | 6.02 | **2.76** | 7.04 | **6.15** | **1.62** | 3.75 | 6.09 | 3.91 | 2.81 | 5.22 | 4.68 | 4.5 |
| L20 [3] | **1.53** | 0.67 | **3.27** | **2.49** | 4.44 | **1.82** | 9.14 | 2.04 | **3.44** | **3.86** | 1.94 | **1.01** | 2.80 | 5.90 | **3.17** |
| Ours | 1.87 | **0.39** | 3.67 | 6.58 | 6.35 | 2.72 | 6.43 | 5.71 | 3.87 | 11.39 | 4.31 | 2.72 | **2.34** | **2.90** | 4.37 |

---

[1]LUCES is licensed under the Apache License, Version 2.0.

L17 [2]

I18 [1]

Q18 [7]

S20 [8]

L20 [3]

Ours

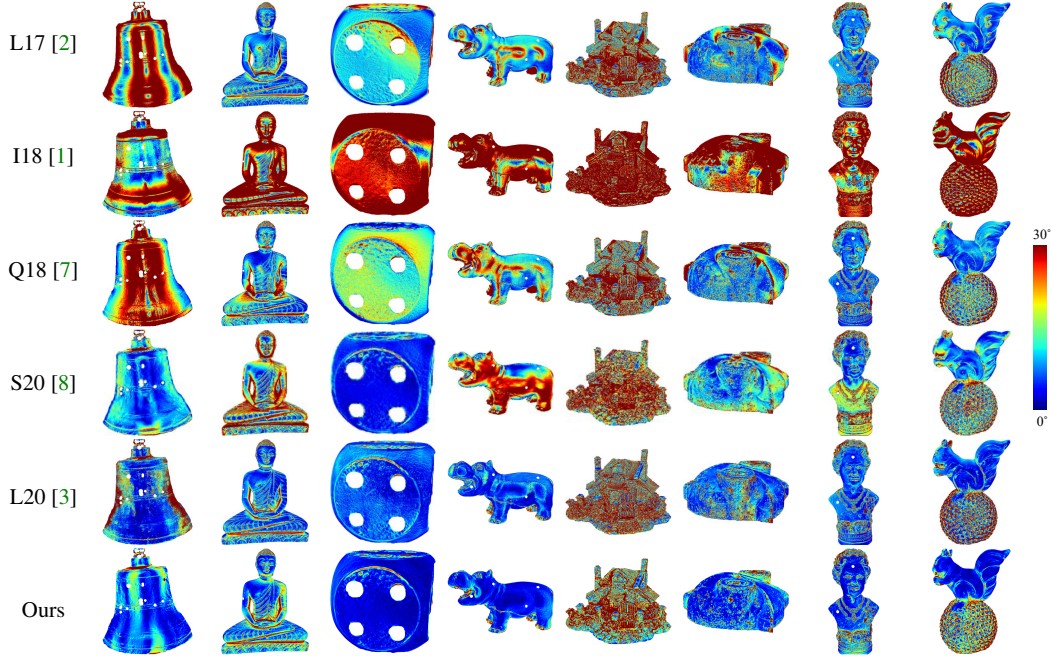

Figure S11: Qualitative comparison with other near-field PS baselines.

| Input | Novel View | Relighting 1 | Relighting 2 | Envmap 1 | Envmap 2 | Edit 1 | Edit 2 |

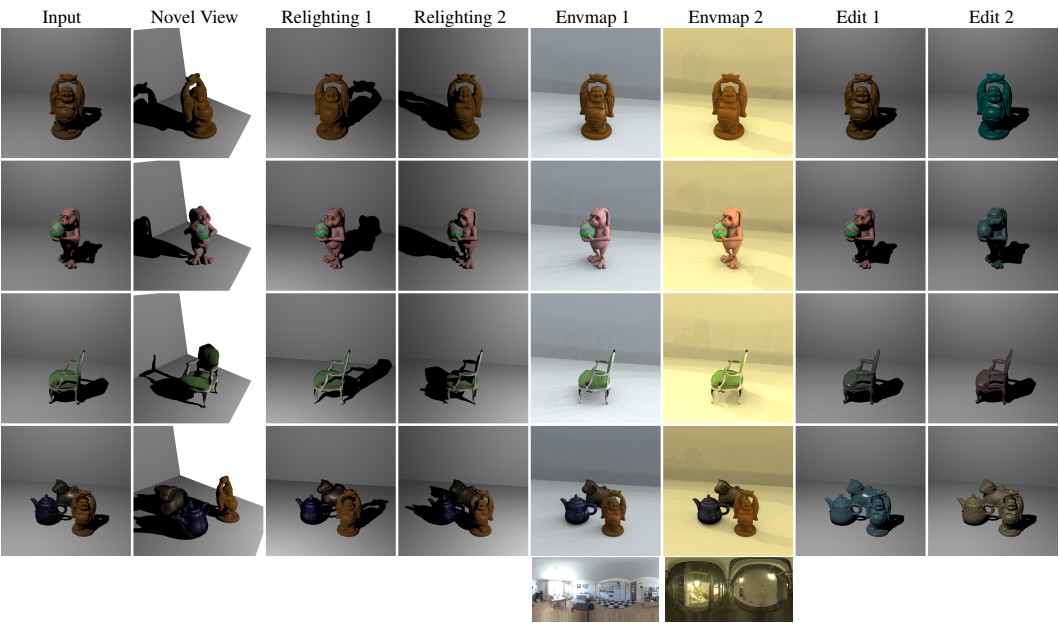

Figure S12: Results for novel-view rendering, relighting, and material editing.

# F More Training Details for the Real Scenes

Since there may exist ambient light in the captured images, we adopt a simple strategy to model the ambient light to stabilize the optimization process. Specifically, we assume the color changes of the observed pixels caused by the ambient light are the product of the predicted albedos and a constant ambient light $A$. we empirically set the constant value $A$ to be a small value as 0.13. The final output (for both $\boldsymbol{C}_v$ and $\boldsymbol{C}_s$) then becomes

$$\boldsymbol{C}_A(\boldsymbol{r}) = \boldsymbol{C}(\boldsymbol{r}) + \rho_d \cdot A. \tag{1}$$

# G Applications

By modeling the scene with a neural reflectance field, our method can disentangle shape, reflectance, and lights. As a result, our method enables applications like novel-view rendering, relighting, and material editing. Figure S12 showcases the results of novel-view rendering, relighting with point light sources and environment map, and material editing. We can see that our method produced visually pleasing rendering and editing results.

# H More Discussions

**BRDF Reconstruction for the Invisible Surface** Our experiments show that the proposed method can utilize shadow information to constrain the shape of the invisible regions viewed from monocular camera. However, when the input images cannot provide many cues for BRDF information of the invisible surfaces, the recovered BRDF might be incorrect in some invisible regions (see Fig. S13). In the future, it would be interesting to utilize sophisticated smoothness regularization or data-driven priors to improve the reflectance estimation in the invisible regions.

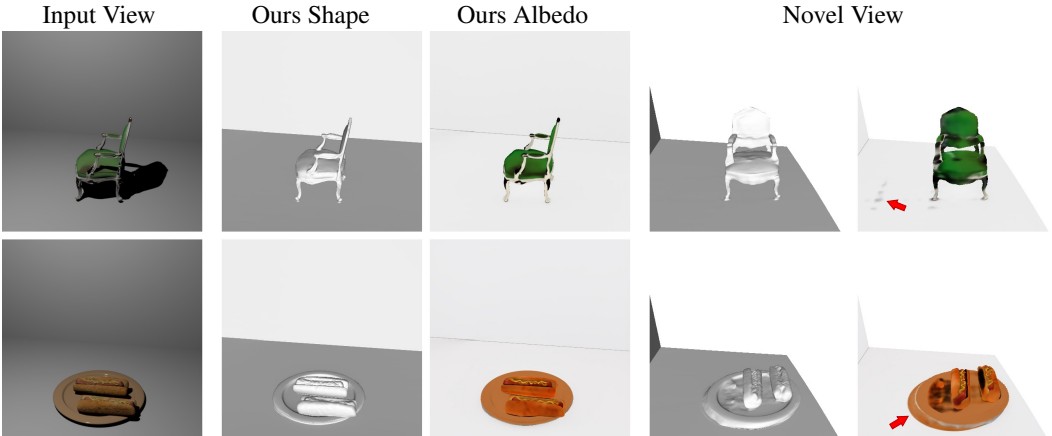

Figure S13: Shape and albedo estimation of our method. The reconstructed albedo in the invisible regions (seen from the camera view) might contain artifacts and noise (as pointed out by the red arrows).

**Potential Negative Societal Impact** Our work can reconstruct the complete shape of a scene from single-view images captured different point lights. This method might be extended to reconstruct invisible regions of a scene from single-view observations, which might cause privacy issues in some situations.