# OpenReview forum: "S$^3$-NeRF: Neural Reflectance Field from Shading and Shadow under a Single Viewpoint"
_NeurIPS.cc/2022/Conference — NeurIPS 2022 Accept_

### Official Review · Reviewer_nudb · 2022-07-08

**Rating:** 8
**Confidence:** 4
**Soundness:** 4 excellent
**Presentation:** 4 excellent
**Contribution:** 4 excellent

**Summary:**

This paper proposed a single-view scene reconstruction method under different point light conditions based on the neural reflectance field. It uses the inverse-square law and BRDF to model the shading and root finding to model the shadow. By explicitly utilizing both shading and shadow cues, the model can reconstruct the scene from single view, even for some invisible regions.
The authors evaluated the model with different objects and showed that the method can well reconstruct the geometry with images from single view but different point lights.


**Questions:**

How’s the normal smoothness loss affect your method? Is that similar to UNISURF? From your result, especially on LEGO, the loss of detail of studs and track seems very serious. Is that caused by normal smoothness loss or by more complicated light reflection at those areas?


**Limitations:**

Yes, the authors addressed different aspect of limitations of the work in the paper and supplementary materials. They also pointed out this method can recover some non-directly seen area, that may address some privacy issues.

**Strengths And Weaknesses:**

### Strengths

A novel NeRF-based method that faithfully reconstructs a scene from a fixed viewpoint with multiple single point light conditions by explicitly modeling the shading and shadow. By carefully modeling the single point light shading by inverse-square law and BRDF, and shadow by root-finding. This method gives an excellent result.
The paper clearly illustrated the method in detail. The authors compared their method with other methods quantitatively and qualitatively. They also evaluated their model with multiple occluding objects, showing it can reconstruct the unseen area with shading and shadow cues.

### Weakness

The authors studied the contribution of different components (shading and shadow) of their method. But the impact of different loss terms was not included. Specially, at L171, how the value of $\alpha$ for normal smoothness loss is found is not illustrated.

---

> ### Author Response · Authors · 2022-08-02
> **Response to Reviewer #nudb**
>
>
> Thank you for the positive comments and insightful suggestions.
>
> **_Q1"The authors studied the contribution of different components (shading and shadow) of their method. But the impact of different loss terms was not included. Specially, at L171, how the value of  for normal smoothness loss is found is not illustrated."_**
> **A1:** Following UNISURF, we apply the normal smoothness loss to promote the surface smoothness. We adopt the same loss weight as in UNISURF [38].
>
>
> **_Q2: "How’s the normal smoothness loss affect your method? Is that similar to UNISURF? From your result, especially on LEGO, the loss of detail of studs and track seems very serious. Is that caused by normal smoothness loss or by more complicated light reflection at those areas?"_**
> **A2:** Thanks for the insightful comment. The adopted normal smoothness loss is similar to UNISURF. Remember that the surfaces in the invisible region are primarily constrained by the shadow information, which is less constrained than the visible surface due to the lack of shading information. To further study the impact of the normal smoothness loss, we did an ablation study on the loss term. Results in https://sites.google.com/view/ps3d (Sec. H) show that imposing the normal smoothness loss is helpful to reduce the artifacts in the invisible regions.
>
> We experimentally verified that the loss of details in LEGO is not caused by normal smoothness loss. We think it might be caused by two reasons. First, LEGO contains many complicated thin structures and the object region is relatively small in our data (in a region of about 200X200 pixels). Second, our method only observes the object from a single view, making thin structures less distinguishable. We will consider improving our method for thin structure modeling in the future.

---

> ### Author Response · Authors · 2022-08-08
> **Look forward to your further feedback!**
>
> Thanks again for the positive comments and insightful suggestions.
>
> We have added the analysis of normal smoothness loss to the supplementary materials (Sec. G). Besides, we also included the experiments on the real data in the main paper in Sec. 4.3.
>
> We also release the data of all real scenes and one synthetic scene, as well as the demo evaluation code in https://github.com/neuralps3d/neuralps3d. The full code and datasets will be released upon acceptance.
>
> We sincerely look forward to your further feedback. Thanks!

---

### Official Review · Reviewer_bpE7 · 2022-07-09

**Rating:** 4
**Confidence:** 4
**Soundness:** 3 good
**Presentation:** 3 good
**Contribution:** 2 fair

**Summary:**

This paper is about using neural radiance field to reconstruction the geometry of an object lit by varying directional light sources, leveraging cues of cast shadow. The optimization uses an efficient way to trace rays for shadow computation.

**Questions:**

- to make the setup slightly more practical, how well does the method perform when shadow is softer, i.e., slightly larger light source, or small area lights?

- line 149: has the proposed shadow generation method been validated over direct regression method using an MLP?

- fig 4: should also show the nearest neighbor training samples

**Limitations:**

The calibrated lighting is more demanding than calibrated camera pose, which can be easily acquired, even though not entirely accurate, on most modern smartphones. The lighting needs to be real directional light source, which is difficult to acquire in real-world scenarios.

It would be also valuable to discuss potential applications for the proposed setup and algorithm.

**Strengths And Weaknesses:**

The setup is interesting as in using shadow cues to reconstruct 3D geometry when multi-view captures are not available. The scope of the paper is quite limited, in terms of the problem setup and the practicality. The setup is demanding -- real directional light plus known lighting direction. This highly controlled lighting setup seems to only be feasible in a laboratory or a light stage setting. This may be the reason the paper only shows toy synthetic examples. Another 'contrived' setup is in the simple scene / background. Cast shadow depends on the object geometry, the lighting, and also the surface the shadow is cast on. The geometry of the surface and the geometry of the object can be entangled to cause ambiguity. This is more severe in real-world examples. The task can be unnecessarily (due to such setup being unlikely in real-world scenarios) challenging and ambiguous without any priors on the object geometry. The reconstructed geometry is not on par with multi-view settings, and degrades drastically as the number of lights decreases. My biggest concern about this paper is it makes the reconstruction task unnecessarily challenging and impractical.

---

> ### Author Response · Authors · 2022-08-02
> **Response to Reviewer #bpE7   [part 1]**
>
>
> Thank you for the constructive comments.
>
> **_Q1: The scope of the paper is quite limited, in terms of the problem setup and the practicality. The setup is demanding -- real directional light plus known lighting direction. This highly controlled lighting setup seems to only be feasible in a laboratory or a light stage setting._**
> **A1:** Note that our method assumes a near-point light, instead of directional light. It is true that the setup of photometric stereo is demanding in the past. However, recent deep lighting calibration methods can achieve robust calibration results by simply using the captured images of the target scene without any calibration tools [A,B], which significantly reduces the complexity of the capturing setup.
> In our experiments on real data (see Sec. A in https://sites.google.com/view/ps3d), we simply use a fixed camera and a handheld cellphone flashlight as the light source. We use SDPS-Net [A] for initial lighting calibration, which is then jointly optimized with the model. Given only weakly calibrated lightings, our method achieves good reconstruction results for normal and shape on this challenging real dataset, clearly demonstrating the practicality of our method.
> [A] Self-calibrating deep photometric stereo networks, CVPR 2019
> [B] Self-calibrating Photometric Stereo by Neural Inverse Rendering, ECCV 2022
>
> **_Q2: "Cast shadow depends on the object geometry, the lighting, and also the surface the shadow is cast on. The geometry of the surface and the geometry of the object can be entangled to cause ambiguity. This is more severe in real-world examples."_**
> **A2:** We agree that the appearance of cast shadow also depends on the surface that shadow is cast on (denoted as background for simplicity). But note that the background geometry itself is also constrained by the shading cues that exist in the multi-light images, which will not introduce ambiguity to the problem. In fact, our results on scenes with multiple objects provide strong evidence for this argument (Figure 7 of the paper). We can treat one of the objects as the foreground, and the rest as the background, resulting in a highly-irregular "background". The results show that our method can faithfully reconstruct the object as well as the "background".
>
> **_Q3: The reconstructed geometry is not on par with multi-view settings, and degrades drastically as the number of lights decreases. My biggest concern about this paper is it makes the reconstruction task unnecessarily challenging and impractical._**
> **A3:** Note that the goal of this work is not to substitute the multi-view stereo methods. Instead, as a PS-based method, our work greatly advances the near-field photometric stereo methods (from 2.5D to 3D reconstruction). This is complementary to existing multi-view methods while only single-view images are required.
> It is obvious that the performance of our method will decrease as the number of lights decreases, but a similar phenomenon also happens in multi-view methods when the number of views decreases. And according to our analysis on light numbers  (Table 4 and Fig. 9 in main paper), our method can achieve comparable results even when only 8 lights are given.
> Our new experiments on real data show that the full shape of an object can be reconstructed from single-view images captured by a fixed camera and a moving handheld cellphone flashlight under a **casual** capturing setup, clearly demonstrating the practicality of our method.
>
>
> **_Q4: "to make the setup slightly more practical, how well does the method perform when shadow is softer, i.e., slightly larger light source, or small area lights?"_**
> **A4:** Thanks for the insightful comment. To analyze the effect of soft shadow caused by a larger light source, we tested our method on data rendered using light sources with different scales (i.e., a sphere with a radius of 1/50, 1/25, or 1/10 of the object size). Results in https://sites.google.com/view/ps3d (Sec. D) show that our method is robust to larger light sources (e.g., 1/50 and 1/25). We also observe that when the light source size is considerably large (e.g., 1/10), the results in the object boundary will decrease because of the heavy soft shadow. Note that this is not a problem in practice as it is very easy to find a point light source whose size is smaller than 1/25 of the object size (e.g., the cellphone flashlight).

---

> ### Author Response · Authors · 2022-08-02
> **Response to Reviewer #bpE7 [part 2]**
>
>
> **_Q5: "line 149: has the proposed shadow generation method been validated over direct regression method using an MLP?"_**
> **A5:** Thanks for the suggestion. We trained a variant model replacing the ray-marching visibility computation with a direct visibility MLP. Results in https://sites.google.com/view/ps3d (Sec. E) show that simply regressing the visibility produces worse results, as this MLP cannot regularize the occupancy field. In contrast, our method performs ray-marching in the occupancy field to render shadow, providing strong constraints for the occupancy field.
>
> **_Q6: "fig 4: should also show the nearest neighbor training samples"_**
> **A6:** Thanks for the suggestion. Results in https://sites.google.com/view/ps3d (Sec. G) show the nearest training samples, and error maps of the nearest sample and our results. We can see that our method can accurately render cast shadows under novel lights. We will include the nearest training samples in Figure 4 in the revised version.
>
>
> **_Q7: "The calibrated lighting is more demanding than calibrated camera pose... The lighting needs to be real directional light source, which is difficult to acquire in real-world scenarios."_**
> **A7:** Please kindly refer to the first response.
>
>
> **_Q8: It would be also valuable to discuss potential applications for the proposed setup and algorithm._**
> **A8:** First, as a photometric stereo method, our method can be applied to any applications that require photometric stereo technique. Second, our method can reconstruct the full shape and BRDFs of the object from single-view images captured under different lightings, and supports novel-view rendering and relighting. It is particularly helpful in scenarios where the camera cannot be moved (e.g., the surveillance camera). Last, the proposed algorithm (e.g., the efficient shadow modeling strategy) can potentially be integrated with multi-view methods for better shape reconstruction.

---

> ### Author Response · Authors · 2022-08-10
> **Repost of Invisible Responses**
>
> Dear Reviewer bpE7,
>
> We just noticed that our previous responses posted under your Acknowledgement thread (titled “Author Rebuttal Acknowledgement by Paper1398 Reviewer bpE7”) are invisible to you and other reviewers. We therefore attach our responses again under this original thread for your reference. Sorry for the inconvenience caused. Thanks!
>
> &nbsp;
>
> --------
> &nbsp;
>
>
> Thank you very much for your previous comments that helped us improve this
> manuscript. We have revised the paper as suggested (highlighted in blue font). The following summarizes the modifications.
> - We have included the discussion for the demands of our setup and the challenges that might occur in a more complicated scene in limitations in Sec. 5.
> - We have added the nearest input image in Fig. 4 of the main paper.
> - We have included experiments on the real data in the main paper in Sec. 4.3.
> - We have added other experiments to the supplementary materials.
> - Implementation details are moved to the supplementary materials (Sec. C) due to page limitation.
>
> We would like to emphasize again that our method tackles the problem of photometric stereo [A], and the setup we proposed is already more practical than existing photometric stereo methods [B,C]. Thus, it might be **unfair** to simply consider our work "The task can be unnecessarily challenging and ambiguous".
>
> We have released the data of all real scenes and one synthetic scene, as well as the demo evaluation code in https://github.com/neuralps3d/neuralps3d. The full code and datasets will be released upon acceptance.
>
> We hope that our response has addressed your concerns and turned you to be positive about the paper. Please feel free to let us know if you have any further concerns or comments. Thanks!
>
> [A] A Benchmark Dataset and Evaluation for Non-Lambertian and Uncalibrated Photometric Stereo, TPAMI 2019
> [B] GPS-Net: Graph-based Photometric Stereo Network, NeurIPS 2020
> [C] Neural Reflectance for Shape Recovery with Shadow Handling, CVPR 2022

---

### Official Review · Reviewer_H2aV · 2022-07-10

**Rating:** 6
**Confidence:** 5
**Soundness:** 3 good
**Presentation:** 3 good
**Contribution:** 2 fair

**Summary:**

This pape proposed a method that aims at solving the near-field photometric stereo via neural reflectance field representation. The core insight of this paper is to use both the shading and cast-shadows of the view as the cues to recover both the visible and invisible parts of a scene.
The paper adopts the NeRF-like neural field representation to represent the occupancy field, reflectance field of a scene. They then use lighting, BRDF modeling and shadow rendering to reconstruct the observed images. The networks are jointly optimized via volume rendering loss, smoothness loss and color loss.

**Questions:**

Please see the quesitons in the Weaknesses part.

**Limitations:**

- Limitations are well discussed in the paper.
- Suggestions: It will be great to isolate the invisible region of the scene and show quantitatively how the proposed shadow rendering improve the accuracy in these regions.

**Strengths And Weaknesses:**

Strengths
- The proposed method can estimate the geometry of the invisible region of the scene from a single-view point. The paper is interesting in estimating invisible regions from the shadows. And unlike previous shadow-based methods, this paper doesn't require explicitly shadow detection as input.
- The method presents state-of-the-art geometry reconstruction results compare to prior near-field photometric stereo methods.
Each component of the method, such as shadow and shading, is validated in the ablation studies.
- The paper is well writen and easy to follow.

Weaknesses
- Each part of the proposed method is not novel. The method consists of three major parts: neural reflectance field representation follows [38] in occupancy field estimation and normal computation; the color and BRDF modeling follow [15,24] to use the sphere gaussian bases, SG weights and diffuse albedo; the shadow computation is slightly different to [48], but it is similar to [24] where they both check the visibility of the surface point. Hence, in the proposed method and rendering procedure of this paper, the novelty is very limited.

- It is suggested to evaluate this method in more challenging real and synthetic datasets. The method was only tested on two datasets: a simple synthetic dataset and a real dataset. For the synthetic dataset: the background is clean and uniform; how will this paper perform in backgrounds with more textures?

- I am curious to see how the number of lights affects the accuracy of the geometry in the invisible part of the scene. The only cues for invisible regions are the casted shadows on the ground. If lights fail to illuminate the invisible regions from different angles, how will the accuracy be affected?

---

> ### Author Response · Authors · 2022-08-02
> **Response to Reviewer #H2aV**
>
>
> Thank you for the positive comments and insightful suggestions.
>
> **_Q1: "Each part of the proposed method is not novel. The method consists of three major parts: neural reflectance field representation follows [38]...the color and BRDF modeling follow [15,24]...the shadow computation is slightly different to [48], but it is similar to [24] where they both check the visibility of the surface point._**
> **A1:** The main contribution of our method is a NeRF-based framework for near-field photometric stereo which can reconstruct the *full* 3D shape of the object by jointly utilizing shading and shadow cues. To the best of our knowledge, we are the first near-field photometric stereo method that can reconstruct the invisible part of the scene. The technical novelty includes "an online shadow modeling strategy for efficient shadow modeling" and "combining volume and surface rendering loss for better shape regularization". We agree that the first two major parts are not very novel. However, our shadow modeling is largely different from [24], as [24] assumes a directional light and uses a depth map for shadow computation. In contrast, we assume a near-field point light and use occupancy field for shadow computation.
>
> **_Q2: "It is suggested to evaluate this method in more challenging real and synthetic datasets."_**
> **A2:** Thanks for the good suggestion. To further evaluate the robustness of our method, we additionally tested our method on a synthetic dataset with more complicated backgrounds, and a challenging real dataset captured by ourselves (see Sec. A & B in https://sites.google.com/view/ps3d). The results show that our method is robust to backgrounds with different lightness and textures, and our method can achieve good results on real data even in a weakly calibrated setup (i.e., the lights are initialized as the prediction of a lighting estimation method SDPS-Net [7] and then jointly optimized with the model).
>
> **_Q3: "I am curious to see how the number of lights affects the accuracy of the geometry in the invisible part of the scene...If lights fail to illuminate the invisible regions from different angles, how will the accuracy be affected?_**
> **A3:** We have discussed the effect of how the number of lights and the range of light distributions on the shape reconstruction in the paper. The normal MAE, depth L1, and side-view shape can be found in Table 4, Table 5, and Figure 9 of the paper. As it is not easy to isolate the invisible part given different objects have different shapes, we instead report the chamfer distance between the reconstructed and ground-truth meshes, which can quantify the full shape reconstruction. Results in https://sites.google.com/view/ps3d (Sec. F) show that the shape accuracy will improve given more lights, and our method is able to achieve robust results given 8 input lights. When the light distribution becomes narrow (small), the shape accuracy will decrease.

---

> ### Author Response · Authors · 2022-08-08
> **Look forward to your further feedback!**
>
> Thanks again for the overall positive comments and insightful suggestions.
>
> We have added the analysis of complicated background (Sec. E.2) and effect of light distribution on invisible regions (Sec. E.6) to the supplementary materials. Besides, we also included the experiments on the real data in the main paper in Sec. 4.3.
>
> We also release the data of all real scenes and one synthetic scene, as well as the demo evaluation code in https://github.com/neuralps3d/neuralps3d. The full code and datasets will be released upon acceptance.
>
> We sincerely look forward to your further feedback. Thanks!

---

### Official Review · Reviewer_13bh · 2022-07-11

**Rating:** 5
**Confidence:** 4
**Soundness:** 2 fair
**Presentation:** 2 fair
**Contribution:** 2 fair

**Summary:**

This paper proposed a promising framework for near-field photometric stereo utilizing volume and surface rendering. The proposed method is tested on a synthesis dataset and a real dataset. Their results have reached state-of-the-art.


**Questions:**

Except for the problems listed above, I wonder how well the proposed method can perform under a more complicated background.


**Limitations:**

The authors have addressed parts of the limitations and potential negative social  impact of their work. However, the impacts of the background's complexity and the shadow's visibility are not stated.


**Strengths And Weaknesses:**

---Originality---
1) The overall idea shows the originality, especially in how to combine surface and volume rendering.
2) Imposing the shadow's constraint on the occupancy field in the 3D space is enlighting.
---Quality---
1) Eq.8: the author used $C_v$ to denote the volume-rendered image and $C_s$ to indicate surface rendered image. However, no formula illustrates how $C_v$ is calculated, which  confuses reading.
2) Figure 2: two "MLP" blocks are shown in two separate branches, which makes me think there are two different MLPs estimating occupancy along the surface-to-light segment and the camera ray. However, according to Eq.6, the MLP for shadow handling is identical to the MLP for the occupancy field. Please try to redraw this part.
---Clarity---
1) Figure 2's caption, lines 2-3: How the points along the rays are sampled is unclear to me. The author mentions to sample $N_V$ points around the surface and $N_L$ points on the surface-to-light segment. I assume $N_L$ points are sampled uniformly, but without knowing the scene's scale, it is unclear how to ensure the sampled points along the camera ray are "around the surface."
2) Although the author tries to explain the necessity of the shadow clues for surface reconstruction in Section 4.4 and Section B in the supplementary material, it is still unclear to me why the shadow becomes a strong constraint on the model that can substitute the constraint from multi-view photo-consistency. There are two kinds of cast shadows in the synthesis dataset, including the foreground shadow on the object, and the background shadow. According to the results in section F, table S2, and section 4.2 table 2, the visibility of the background shadow is very important for depth estimation. The author should analyze the contribution of two different kinds of shadows separately.
3) According to the presenting results, the proposed method has reached an ideal visual and quantitative result in their synthetic dataset. Even the background is well reconstructed with little artifacts. However, the author didn't directly explain why their method is not sensitive to the background (it is almost perfect) and can reconstruct an accurate background simultaneously. (e.g., UNISURF [A] applies NeRF++ [B] for the complex background and assumes black for the simple background. However, their reconstructed background still contains visible artifacts).

[A] Michael Oechsle, Songyou Peng, and Andreas Geiger. UNISURF: Unifying neural implicit surfaces and radiance fields for multi-view reconstruction. In Proceedings of the IEEE/CVF International Conference on Computer Vision (ICCV). 2021

[B] Kai Zhang, Gernot Riegler, Noah Snavely, and Vladlen Koltun. Nerf++: Analyzing and improving neural radiance fields. arXiv.org, 2020.

---Significance---

1) There is no intuitive explanation for the role of modeling shadow for shape estimation. What is the problem aiming to solve by considering the shadow, the ambiguity, additional clue of global shape information, useful clue for depth estimation, or even light refinement?
2) The evaluation is mainly conducted on the synthetic dataset built by the authors, which is less convincing. Although the paper shows the result of the LUCES dataset in the supplementary material, the depth estimation is not satisfactory (4.39 vs. 3.17 for the state-of-the-art). This evaluation could not provide strong support to the proposed method.
3) The compared methods, NeRF and UNISURF, are implemented in a naive way. It is unclear whether the authors have also sampled points along the light ray in those methods. If not, it can be expected that the reconstruction results are not good because NeRF and UNISURF are mainly applied in multi-view stereo, which is constrained by the multi-view photo consistency that is lacking in single-view photometric stereo. Therefore, using them for comparison is less convincing.
4) As said before, the background shadow's visibility greatly influences the performance. Therefore, this work has lots of pre-setting to the environment (the background is better to be the light color, and the intensity should be bright enough), which will limit the application scope of this method. Moreover, The background's complexity may influence the background shadow's visibility and the shading of the scenes, which further impact the performance of the methods. While the background in the synthesis dataset and LUCES is still an ideal case, there is a lack of such discussions on a more realistic scenario.

---

> ### Author Response · Authors · 2022-08-02
> **Response to Reviewer #13bh   [part 1]**
>
> Thank you for the constructive comments.
>
> **_Q1: No formula illustrates how $C_v$ in Eq.8 is calculated._**
> **A1:** Sorry for the confusion. The $C_v$ mentioned in Eq.8 is actually the $C(r)$ in Eq.7. We first derive volume rendering and later introduce surface rendering in Eq.10. We will clarify it in Sec. 3.4 in the revised version.
>
> **_Q2: In Figure 2, the same "MLP" block is shown in two separate branches._**
> **A2:** Thanks for the good suggestion. We will redraw Figure 2 in the revised version.
>
> **_Q3: "How the points along the rays are sampled is unclear to me"_**
> **A3:** Sorry for the confusion. The points along the camera ray are sampled following the manner of UNISURF [38]. For each ray, we first apply root-finding to locate the surface point, and then define an interval around the surface point to sample points. We use a relatively large interval for point sampling to make sure the full scene is well sampled. The term "around the surface" does not mean "close to the surface". We will update the caption in Figure 2 to avoid confusion.
>
>
> **_Q4: "It is still unclear to me why the shadow becomes a strong constraint on the model that can substitute the constraint from multi-view photo-consistency"_**
> **A4:** This paper tackles **the problem of near-field photometric stereo**, which is a longstanding and important problem in computer vision [46]. Our goal is **not to substitute** the multi-view photo-consistency with shadow constraints. Instead, considering a scenario where single-view images are captured under different point lights, we target utilizing both shading and shadow information for full 3D shape reconstruction. Note that the shadow cues can also be incorporated with the multi-view photo-consistency to improve the shape reconstruction.
>
> **_Q5: Analyze the contributions of cast shadows modeling in foreground and background separately._**
> **A5:** Thanks for the insightful comment. We have conducted extra experiments to analyze the effect of cast shadow modeling in both regions. Specifically, we trained two variant models, one without foreground shadow modeling and the other without background modeling. Results in https://sites.google.com/view/ps3d (Sec. C) show that modeling cast shadow in both regions is important, as disabling either one of them leads to decreased accuracy.
>
> **_Q6: "Even the background is well reconstructed with little artifacts. However, the author didn't directly explain why their method is not sensitive to the background (it is almost perfect) and can reconstruct an accurate background simultaneously."_**
> **A6:** We do **not** claim that our method is better at reconstructing the background than other methods. There are possible two reasons why we got clean background results in the paper. First, the background in our synthetic dataset has a uniform texture with a regular shape, which is easier to recover. Second, our method makes use of the rich shading information that exists in images illuminated by multiple lights, providing strong regularization for the planar background.

---

> > ### Comment · Reviewer_13bh · 2022-08-07
> > **Post-rebuttal comments from Reviewer 13bh**
> >
> > My major concern about "*how the proposed method contributes to the community*" is adequately addressed by the additional experiments provided in the rebuttal. These additional results are crucial, and the authors should add them to the main paper as primary supporting results.
> >
> > Besides, I would be more convinced if the author could **provide the code and data** for me to reproduce the results. I hope the authors could **release** their data and code if the paper got accepted as they promised.  I was the reviewer of an ICCV 2021 PS paper, and the authors of that paper promised in their rebuttal to release their code once their paper was accepted. However, as far as I know, that paper's code is still unavailable.
> >
> > At this time, I am leaning toward increasing my rating to acceptance.

---

> > > ### Author Response · Authors · 2022-08-08
> > > **Response to Reviewer #13bh : Release of Code and Data**
> > >
> > > Thanks for your positive feedback, which has led to a substantial improvement of our paper's quality! According to your comments and suggestions, we have added the additional results in the revised paper (highlighted in blue font). The following summarizes the modifications.
> > > - We have updated Fig. 2 and its caption and clarified the $C_v$ in Sec. 3.4.
> > > - We have included the experiments on real data in the main paper in Sec. 4.3.
> > > - We have added other additional experiments in the supplementary materials.
> > > - Due to the page limit, we moved the implementation details to supplementary materials (Sec. C).
> > >
> > > As promised in the abstract of our paper, we will release the code and dataset upon acceptance. In this stage, to help the reviewers better understand our method, we release the data of all real scenes and one synthetic scene, as well as the demo evaluation code in https://github.com/neuralps3d/neuralps3d. The full code and datasets will be released upon acceptance. Please kindly note that all the reviews and discussions on the OpenReview system will be made **public** after the paper gets accepted. We will keep our commitment.
> > >
> > > Please feel free to let us know if you have any further concerns or comments. Thanks!

---

> ### Author Response · Authors · 2022-08-02
> **Response to Reviewer #13bh   [part 2]**
>
>
> **_Q7: "There is no intuitive explanation for the role of modeling shadow for shape estimation. What is the problem aiming to solve by considering the shadow, the ambiguity, additional clue of global shape information, useful clue for depth estimation, or even light refinement"_**
> **A7**: Our method explicitly makes use of this additional shadow cue for full shape reconstruction. Note that our method considers the classic problem of near-field photometric stereo, where the shading and shadow are two dominant cues for shape recovery. The shadows can provide strong information for the invisible region, which is analogy to shape from silhouette. Considering the light source as the camera viewpoint, shadow is formed when light rays are blocked by objects. If we view from the light source, the contour of the shadow is the same as the silhouette of the object on the image plane (light projects the contour of objects onto the scene, while objects are projected back onto the image plane).
>
> **_Q8: "The evaluation is mainly conducted on the synthetic dataset built by the authors, which is less convincing. Although the paper shows the result of the LUCES dataset in the supplementary material, the depth estimation is not satisfactory (4.39 vs. 3.17 for the state-of-the-art). "_**
> **A8:** Note that the experiment on LUCES does not aim to achieve state-of-the-art depth estimation performance but shows that our method is able to work on the **real data** for completeness. In fact, when testing on the LUCES dataset, our method degenerates to the *"Ours w/o shadow"* baseline, which is much worse than the full model (see Table 3 and Figure 6 of the paper). As mentioned in Line 47 of the supplementary material, the LUCES dataset is **not** suitable to evaluate the full potential of our method, as the shadow and shading information of the background regions cannot be observed.
> To further evaluate our method, we evaluate our method on a real dataset captured by ourselves. Results in https://sites.google.com/view/ps3d (Sec. A) show that our method achieves **good normal** and **shape reconstruction** results on the **real** dataset given only roughly calibrated lightings, clearly demonstrating the effectiveness of our method.
>
> **_Q9: "The compared methods, NeRF and UNISURF, are implemented in a naive way. It is unclear whether the authors have also sampled points along the light ray in those methods. If not, it can be expected that the reconstruction results are not good because NeRF and UNISURF are mainly applied in multi-view stereo..."_**
> **A9**: We are not about to show the superiority over NeRF/UNISURF on single view reconstruction. Instead, we would like to demonstrate that directly applying these neural rendering methods or naively conditioning light on the radiance field does not work for single view reconstruction. Note that we compared three types of methods in the experiments, including the single-image depth estimation, near-field photometric stereo methods, and NeRF-based methods. We did not include the explicit shadow computation (mentioned as "sample points along the light ray") for the NeRF-Based methods, as the shadow modeling is one of our main technical contributions. In fact, the "UNISURF+shadow" method is similar to the baseline method "Ours w/o shading" in the ablation study (see Line 228 of the paper). Table 3 of the paper shows that simply integrating UNISURF and shadow modeling cannot produce good results, e.g., MAE comparison for the CHAIR object is "Ours w/o shading" (32.49) vs. "Ours" (1.93).
>
>
> **_Q10: This work has lots of pre-setting to the environment (the background color and the background complexity), which will limit the application scope of this method._**
> **_Q11: How well the proposed method can perform under a more complicated background._**
> **A10-11**: To further evaluate the robustness of our method on more complicated backgrounds, we evaluated it on four scenes rendered with different backgrounds, including two uniform color backgrounds with different lightness (denoted as 'Light' and 'Dark') and two textured backgrounds. Results in https://sites.google.com/view/ps3d (Sec. B) show that our method is robust to backgrounds with different lightness and textures. Moreover, our new real-world experiments show that the capturing setup can be very simple.

---

### Author Response · Authors · 2022-08-02
**General Response: Contributions and New Experiments**

We sincerely appreciate all reviewers’ and ACs’ time and efforts in reviewing our paper. We thank all the reviewers for their recognization of our work on the following aspects.
* **Problem Setup.** *"interesting setup"* [H2aV,bpE7];
* **Model.** *"a promising framework for near-field photometric stereo"* [13bh], *"shows the originality"* [13bh];
* **Experiments.** *"state-of-the-art performance"* [13bh,H2aV,nudb];
* **Wrting.** *"well-written paper"* [H2aV,nudb].

And we also thank all reviewers for their insightful and constructive suggestions, which help a lot in further improving our paper. In addition to the pointwise responses below, we clarify our idea and contribution, and summarize the new experiments suggested by the reviewers.

**Idea and Contribution**

In this work, we propose a NeRF-based method for near-field photometric stereo (PS). By explicitly utilizing single-view shading and shadow cues, our method is able to reconstruct **the full 3D shape** of the object, including the **visible** and **invisible** regions. It is not possible by existing PS methods as they can only recover a **2.5D scene** representation (i.e., normal or depth map) to describe the **visible** surface.
Note that our method is based on PS settings, where multi-view information is not available. And the goal of our method is not to substitute the multi-view stereo methods. Instead, we propose to utilize both shading and shadow information in the scene to improve the current PS methods (from 2.5D to 3D shape reconstruction). Our work is complementary to existing multi-view methods while only single-view images are required. Moreover, our idea of jointly modeling shading and shadow cues is also potentially beneficial for multi-view reconstruction.

**New Experiments**

To address the reviewers' questions and support our responses, we conduct the following experiments and put the results in the link: https://sites.google.com/view/ps3d.
- Results on real data with casual imaging setup [13bh,H2aV,bpE7].
- Results on more complicated synthetic data [13bh,H2aV,bpE7].
- Effects of shadow modeling in foreground and background [13bh].
- Results on scenes illuminated by area light [bpE7].
- Replacing ray-marching shadow computation with direct MLP regression [bpE7].
- Effects of the surface normal smoothness loss [nudb].

**Details for the Experiment on Real Data**
- Casual capturing setting
We captured a real dataset using a fixed camera (the focal length is 28mm) and a handheld cellphone flashlight (see Sec. A in https://sites.google.com/view/ps3d). The object is put on the table and close to the wall. We turned off all the environmental light sources and only kept the flashlight on, which was randomly moved around to capture images illuminated under different light conditions. The captured dataset consists of 3 objects (around 70 images for each object).
- Lighting calibration
Our setup **does not** require manual calibration of lights. Instead, we applied the state-of-the-art self-calibrated photometric stereo network (SDPS-Net [7]) for light direction initialization, and roughly measured the camera-object distance as initialization of light-object distance. After initialization, the position and direction of lights are jointly optimized with the shape and BRDF during training.
- Results
We tested our method on three objects and show their rerendered results as well as shape reconstruction results in https://sites.google.com/view/ps3d (Sec. A). Even with this **casual capturing** setup and **uncalibrated** lights, our method achieves satisfactory results in full 3D shape reconstruction.

---

### Author Response · Authors · 2022-08-06
**Thanks for all your comments and look forward to post-rebuttal feedbacks!**

Dear AC and all reviewers:

Thanks again for all of your constructive suggestions, which have helped us improved the quality and clarity of the paper!

Since the discussion phase has started for over three days, we have not heard any post-rebuttal response yet.

Please don’t hesitate to let us know if there are any additional clarifications or experiments that we can offer, as we would love to convince you of the merits of the paper. We appreciate your suggestions. Thanks!

---

### Author Response · Authors · 2022-08-08
**General Response II : Release of Code and Data**

Thanks again for all reviewers' constructive comments. We have modified our paper according to reviewers' suggestions in the revised version, including
- update Fig. 2 and its caption [13bh] and Fig. 4 [bpE7],
- clarify $C_v$ in Sec. 3.4 [13bh],
- include experiments on real data in the main paper in Sec. 4.3 [13bh,H2aV,bpE7],
- include the discussion of the demands of our setup and the challenges that might occur in a more complicated scene in limitations in Sec. 5 [bpE7],
- add other experiments and move the implementation details to supplementary materials (Sec. C),
    - analysis on complicated background (Sec. E.2) [13bh,H2aV,bpE7],
    - analysis on shadow modeling in foreground and background regions (Sec. E.3) [13bh],
    - compare with MLP regression for shadow computation (Sec. E.4) [bpE7],
    - effect of area light (Sec. E.5) [bpE7],
    - effect of light distribution for invisible shape reconstruction (Sec. E.6) [H2aV],
    - effect of normal smoothness loss (Sec. E.7) [nudb],
    - more details of real dataset (Sec. G) [13bh,H2aV,bpE7].


We also release the data of all real scenes and one synthetic scene, as well as the **demo evaluation code** in https://github.com/neuralps3d/neuralps3d. The full code and datasets will be released upon acceptance. We hope our code and dataset can benefit future research in this direction.

Please feel free to let us know if you have any further concerns or comments. Thanks!

---

### Meta-Review · Area_Chair_MAEc · 2022-09-01

**Recommendation:** Accept
**Confidence:** Certain

**Metareview:**

This paper had reviews ranging from borderline reject to strong accept.  The most negative reviewer had concerns about the assumptions in the framework (point light sources), and the loss of accuracy as the number of light sources decreases, but the remaining reviewers were compelled by the ability to hand scenes with lights not at infinity and the integration of the shadow constraints to give constraints on the structures of scene parts not directly viewed.

Overall I agree with the three positive reviewers that this paper considers an interesting variation of the photometric stereo problem with coherent experimental evaluation that shows the contributions of each of the different pieces of their overall system

Therefore I accept this paper.


**Award:**

No

---

### Decision · Program_Chairs · 2022-09-14

Accept